**Nitrate radical oxidation of γ-terpinene: hydroxy nitrate, total organic nitrate, and**
**secondary organic aerosol yields**
Jonathan H. Slade[1*], Chloé de Perre[2], Linda Lee[2], and Paul B. Shepson[1,3]
[1]Department of Chemistry, Purdue University, West Lafayette, IN 47907
[2]Department of Agronomy, Purdue University, West Lafayette, IN 47907
[3]Department of Earth, Atmospheric, and Planetary Sciences, Purdue University, West Lafayette,
IN 47907
[*]Corresponding author: jslade@purdue.edu
**Abstract**

10        Polyolefinic monoterpenes represent a potentially important but understudied source of

organic nitrates (ON) and secondary organic aerosol (SOA) following oxidation due to their high
reactivity and propensity for multi-stage chemistry. Recent modeling work suggests that the
oxidation of polyolefinic γ-terpinene can be the dominant source of nighttime ON in a mixed forest
environment. However, the ON yields, aerosol partitioning behavior, and SOA yields from γ-
terpinene oxidation by the nitrate radical ($NO_3$), an important nighttime oxidant, have not been
determined experimentally. In this work, we present a comprehensive experimental investigation
of the total (gas + particle) ON, hydroxy nitrate, and SOA yields following γ-terpinene oxidation
by $NO_3$. Under dry conditions, the hydroxy nitrate yield = 4(+1/-3)%, total ON yield = 14(+3/-
2)%, and SOA yield ≤ 10% under atmospherically-relevant particle mass loadings, similar to those
for α-pinene + $NO_3$. Using a chemical box model, we show that the measured concentrations of
$NO_2$ and γ-terpinene hydroxy nitrates can be reliably simulated from α-pinene + $NO_3$ chemistry.
This suggests that $NO_3$ addition to either of the two internal double bonds of γ-terpinene primarily
decomposes forming a relatively volatile keto–aldehyde, reconciling the small SOA yield observed
here and for other internal olefinic terpenes. Based on aerosol partitioning analysis and
identification of speciated particle-phase ON applying high-resolution liquid chromatography–
mass spectrometry, we estimate that a significant fraction of the particle-phase ON has the hydroxy
nitrate moiety. This work greatly contributes to our understanding of ON and SOA formation from
polyolefin monoterpene oxidation, which could be important in the northern continental U.S. and
Midwest, where polyolefinic monoterpene emissions are greatest.
**1.  Introduction**
The oxidation of volatile organic compounds (VOCs) is a major pathway in the production
of secondary organic aerosol (SOA), which can represent up to ~60% of the total submicron
aerosol mass, depending on location (Hallquist et al., 2009;Riipinen et al., 2012;Glasius and
Goldstein, 2016). Aerosols impact climate by scattering and absorbing radiation as well as
modifying cloud optical properties, and can adversely affect human health (Stocker et al., 2013).
A large fraction of the total OA budget derives from the oxidation of *biogenic* VOCs (BVOCs),
including isoprene ($C_5H_8$) and monoterpenes ($C_{10}H_{16}$) (Hallquist et al., 2009;Spracklen et al.,
2011). Together, these naturally emitted compounds account for ~60% of the global BVOC budget
(Goldstein and Galbally, 2007;Guenther et al., 1995). In particular, monoterpenes, comprising
~11% of the total global BVOC emissions (Guenther, 2002), represent a viable source of SOA
following oxidation (Griffin et al., 1999;Lee et al., 2006). However, atmospheric models routinely
underestimate the global SOA burden (Kokkola et al., 2014), causing a potential order of
magnitude error when predicting global aerosol forcing (Goldstein and Galbally, 2007), and thus
the sources and mechanisms responsible for SOA formation require further study.
VOC oxidation produces an array of semi-volatile organic aerosol precursors, including
organic nitrates ($RONO_2$), herein referred to as "ON", in the presence of $NO_x$ (i.e., $NO+NO_2$)
(Kroll and Seinfeld, 2008;Rollins et al., 2010b;Rollins et al., 2012;Darnall et al., 1976). By
sequestering $NO_x$, ON can perturb ozone concentrations globally (Squire et al., 2015). Moreover,
as $NO_x$ concentrations are expected to decrease in the future (von Schneidemesser et al., 2015),
ambient concentrations of $NO_x$ and thus $O_3$ will become increasingly sensitive to ON formation
(Tsigaridis and Kanakidou, 2007). Monoterpenes contribute significantly to the formation of ON
and SOA, especially during nighttime in the presence of nitrate radicals ($NO_3$), when isoprene
concentrations are negligible and the photolytic and NO reaction sinks of $NO_3$ are cut off (see Ng
et al. (2017) and references therein). It is estimated that monoterpene oxidation by $NO_3$ may
account for more than half of the monoterpene-derived SOA in the U.S., suggesting that ON is a
dominant SOA precursor (Pye et al., 2015). However, their formation mechanisms and yields
following oxidation by $NO_3$ are not as well constrained as those from OH and $O_3$ oxidation (Hoyle
et al., 2011), and previous studies have focused on the $NO_3$ oxidation of only a few monoterpenes
(Fry et al., 2014), but almost exclusively on mono-olefinic terpenes such as α- and β-pinene (Boyd
et al., 2015;Spittler et al., 2006;Wangberg et al., 1997;Fry et al., 2009;Berkemeier et al., 2016).
An important detail is the relative amount of hydroxy nitrates produced, as the -OH group
contributes greatly to water solubility (Shepson et al., 1996), and uptake into aqueous aerosol
followed by continuing chemistry in the aqueous phase, which can be an important mechanism for
SOA production (Carlton and Turpin, 2013).

A major challenge regarding our understanding of SOA formation from monoterpene

oxidation is that there are several isomers of monoterpenes with very different structural
characteristics that can exhibit very different yields of SOA following oxidation (Fry et al.,
2014;Ziemann and Atkinson, 2012). For example, the SOA mass yield from the $NO_3$ oxidation of
α-pinene, which contains one endocyclic double bond, is ~0% under atmospherically relevant
particle mass loadings, whereas that from β-pinene, which contains one terminal double bond, is
33% under the same experimental conditions (Fry et al., 2014). Limonene, with one tertiary endo-
and one terminal exocyclic double bond, also exhibits relatively larger SOA mass yields following
oxidation by $NO_3$ (Fry et al., 2014;Spittler et al., 2006). Because $NO_3$ oxidation of α-pinene
primarily leads to tertiary peroxy radical formation (Wangberg et al., 1997), the initially formed
alkoxy radical rearranges to a ketone and decomposes the nitrooxy group, releasing $NO_2$ and
forming a keto–aldehyde, which has higher saturation vapor pressure compared to its ON analogue
(Pankow and Asher, 2008). However, decomposition of the nitrate is not exclusive for all tertiary
alkoxy radicals following $NO_3$ oxidation as it may also depend on the structure of the adjacent
bond. Based on structure–activity relationships, a β-alkyl substitution is expected to destabilize the
adjacent bond more than a β-nitrate substitution (Vereecken and Peeters, 2009). In the case of β-
pinene or sabinene, for example, the expected decomposition pathway of the alkoxy radical leaves
the nitrooxy group intact to form a keto–nitrate (Fry et al., 2014). SOA yields have also been shown
to be strongly dependent on the total (gas + particle) yield of ON. Owing to their low saturation
vapor pressures, multifunctional ON such as the hydroxy nitrates are thought to contribute
significantly to SOA formation (Rollins et al., 2010a;Rollins et al., 2010b;Lee et al., 2016), and
have been the focus of several laboratory and field research campaigns including the BEACHON
2011 field study in the Colorado front range (Fry et al., 2013), the BEARPEX 2009 study at the
Blodgett forest site in the Western foothills of the Sierra Nevada (Beaver et al., 2012), the
PROPHET and SOAS field studies in the upper Midwest and southeastern US (Xiong et al.,
2015;Lee et al., 2016;Grossenbacher et al., 2004), and the Focused Isoprene eXperiment at the
California Institute of Technology (FIXCIT) (Nguyen et al., 2014). These ON can rapidly undergo
aqueous-phase processing, especially under acid-catalyzed conditions, to form diols and
organosulfates (Jacobs et al., 2014;Rindelaub et al., 2015;Rindelaub et al., 2016;Surratt et al.,
2008), which not only complicates quantification of organic nitrates in the aerosol phase (Russell
et al., 2011), but affects product saturation vapor pressure and thus aerosol formation, represents
a sink for $NO_x$, and may affect the hygroscopic properties of organic aerosol (Suda et al., 2014).
However, considering there are only a limited number of studies that have specifically investigated
the yield of hydroxy nitrates, namely following OH and $NO_3$ oxidation of isoprene (Chen et al.,
1998;Lockwood et al., 2010;Xiong et al., 2015) and α-pinene (Rindelaub et al., 2015;Wangberg et
al., 1997), further measurements of their yields and role in aerosol formation from the oxidation
of other terpenoids is critical.

In the southeastern U.S., α- and β-pinene tend to dominate monoterpene emissions (Geron

et al., 2000), and their potential for ON and SOA formation are better-studied compared to other
monoterpenes (Ayres et al., 2015;Lee et al., 2016). However, in other regions of the U.S.,
polyolefinic monoterpenes such as terpinene, ocimene, and limonene can be present in much
greater proportions than in the southeastern US, which may be in part due to the relatively smaller
abundance of the α- and β-pinene emitter southern pine, but also more polyolefinic monoterpene
emitters, including *Juniperus scopulorum*, a common cedar and γ-terpinene emitter in the
Midwestern US (Geron et al., 2000). In particular, model simulations suggest that the oxidation of
γ-terpinene, comprising two substituted endocyclic double bonds, can contribute as much as α-
and β-pinene to nighttime organic nitrate production in a mixed northern hardwood forest (Pratt et
al., 2012). Those authors also showed that $NO_3$ reaction with BVOCs is important in the daytime.
However, the ON and SOA yields following $NO_3$ oxidation of γ-terpinene have not been
determined in laboratory studies.

Here we present a comprehensive laboratory investigation of the hydroxy nitrate, total gas

and particle-phase ON, and SOA yields from the $NO_3$ oxidation of γ-terpinene. For the hydroxy
nitrate yield experiments, a surrogate standard compound was synthesized as presented in the
supplemental information of Rindelaub et al. (2016), enabling quantitative determination of its
yield using a chemical ionization mass spectrometer (CIMS). This work contributes to a broader
understanding of SOA formation from the oxidation of polyolefinic monoterpenes, and the role of
$NO_3$ oxidation chemistry in the sequestration of $NO_x$.

**2. Methods**

Yield experiments were conducted in a 5500 L photochemical reaction chamber with

Teflon walls and perfluoroalkoxy (PFA)-coated endplates, in the dark (Chen et al., 1998). Briefly,
the chamber was cleaned by flushing several times with ultra-zero (UZ) air in the presence of ultra-
violet light. Experiments were conducted in a dry atmosphere (relative humidity < 1%) and at
ambient temperature (~295 K). A total of 13 independent yield experiments were conducted over
a range of initial γ-terpinene concentrations in the presence of $N_2O_5$ with and without $(NH_4)_2SO_4$
seed particles. $N_2O_5$ was produced in a dried glass vessel and crystallized at 195 K in a custom-
made glass trap following thermal equilibrium with $NO_2$ and $O_3$, as indicated in reactions (1) and
(2) below.

$O_3$   +   $NO_2$   →   $NO_3$   +   $O_2$              (1)

$NO_3$   +   $NO_2$   ↔   $N_2O_5$                       (2)

First, the BVOC was transferred to the chamber with UZ air via injection through a heated glass
inlet and polytetrafluoroethylene (PTFE) line. For the seeded experiments, $(NH_4)_2SO_4$ particles
were generated by passing an aqueous solution through a commercial atomizer (Model 3076, TSI,
Inc.) and subsequently dried through a diffusion dryer prior to entering the reaction chamber. The
seed particles were polydisperse with a range in the geometric mean diameter, $D_{p,g}$, of 57 nm to
94 nm and geometric standard deviation, $\sigma_g$, of 1.39 to 1.91. Total seed number and mass
concentrations were in the range $0.61\text{-}5.15\times10^4$ cm$^{-3}$ and 8-48 µg m$^{-3}$, respectively, assuming a
seed particle density of 1.7 g cm$^{-3}$. Yield experiments were initiated (time = 0) by injecting $N_2O_5$
into the chamber with a flow of UZ air over the crystalline $N_2O_5$. The reactants were allowed to
mix continuously in the chamber with a fan, and the reaction was terminated when no less than
10% of the γ-terpinene remained to limit secondary particle-phase or heterogeneous $NO_3$
chemistry.
Real-time measurements were made using several instruments: γ-terpinene concentrations
were measured with a gas chromatograph-flame ionization detector (GC-FID; HP-5890 Series II),
which was calibrated using a commercial γ-terpinene standard dissolved in cyclohexane. $NO_2$
concentrations were measured with a custom-built chemiluminescence $NO_x$ analyzer (Lockwood
et al., 2010), and a scanning mobility particle sizer (SMPS; Model 3062, TSI, Inc.) was used to
determine size-resolved particle mass concentrations. No direct concentration measurements of
$NO_3$ were made. The hydroxy nitrates were measured online continuously using an iodide-adduct
chemical ionization mass spectrometer (CIMS) (Xiong et al., 2015;Xiong et al., 2016). To quantify
the production of monoterpene hydroxy nitrates, the CIMS was calibrated with a purified standard
of an α-pinene-derived hydroxy nitrate synthesized in-house via nitrification of α-pinene oxide
(Sigma–Aldrich, 97%) using $Bi(NO_3)_3 \cdot 5H_2O$ (Rindelaub et al., 2016). The concentration of the
purified hydroxy nitrate was verified via two complementary methods: [1]HNMR and FTIR, and the
structure was verified using [13]C-NMR, as presented in the supplementary information of Rindelaub
et al. (2016). The total ON yields and concentration of the standard were determined via FTIR
measurement of the asymmetric –$NO_2$ stretch located at ~1640 cm[-1] using tetrachloroethylene
(Sigma–Aldrich, HPLC grade, ≥99.9%) as the solvent (Rindelaub et al., 2015). We note that the
FTIR approach cannot distinguish mono- from poly-nitrated organics. However, given the
relatively low concentrations of $NO_2$ compared to $O_2$ in the chamber and their rate constants with
alkoxy radicals (Atkinson et al., 1982), first-generation di-nitrates constitute an insignificant
fraction of ON (<0.2%). Second-generation di-nitrates from $NO_3$ reaction at the remaining double
bond on γ-terpinene, however, may account for a maximum of ~10% of the total ON based on the
relative rates of primary to secondary monoterpene oxidation reactions. Thus the uncertainties for
our reported yields include a component from this uncertainty (10%) in the fraction of the nitrates
that are dinitrates. The total gas-phase ON yields were determined with FTIR following the
sampling of chamber air through an annular denuder (URG-200) coated with XAD-4 resin and
extraction from the denuder walls with tetrachloroethylene as in a previous study (Rindelaub et
al., 2015). Aerosol particles were collected on 47 mm PTFE filters (1 μm pore size; ~100%
collection efficiency) housed in a cartridge connected to the denuder exit. The collection efficiency
of the denuder walls for gas-phase organic nitrates was determined to be >98% based on
measurements of the concentration of 2-ethyl-hexyl-nitrate (Sigma–Aldrich, 97%) before and after
the denuder with the GC-FID. The particle transmission efficiency was determined to be >98% by
measuring the number concentration of particles before and after the denuder with the SMPS.

Wall loss and dilution corrections were applied to both the SOA and ON yields accounting

for the time required to sample through the denuder. Following several of the experiments, the
SOA concentration was measured as a function of reaction time with the wall with an average wall
loss rate constant, $k_{wall,SOA} = 9\times10^{-5}$ s$^{-1}$. The gas-phase ON wall loss rate was determined based on
the evolution of the CIMS-derived monoterpene hydroxy nitrate (M=$C_{10}H_{17}NO_4$) signal ([M+I]$^-$;
$m/z$ = 342) following an experiment, in which we obtained $k_{ONg} = 2\times10^{-5}$ s$^{-1}$, as shown in Fig. S1.

Selected filter extracts from two separate chamber experiments were analyzed for their

chemical composition via ultra-performance liquid chromatography electrospray ionization time-
of-flight tandem mass spectrometry (UPLC-ESI-ToF-MS/MS, Sciex 5600+ TripleToF with
Shimadzu 30 series pumps and autosampler) to identify potential ON species in the particle phase
from γ-terpinene oxidation by $NO_3$. The samples were first dried with ultra-high purity nitrogen
and then extracted with a 1:1 v:v solvent mixture of HPLC-grade methanol and 0.1% acetic acid
in nanopure $H_2O$, which has been used successfully as a solvent system for identifying
multifunctional organonitrate and organosulfate species (Surratt et al., 2008).
**3. Results and Discussion**
*3.1. SOA yields*
Mass-dependent SOA yields ($Y_{SOA}$) were derived from both seeded and unseeded
experiments and defined here as the change in aerosol mass concentration ($\Delta M$ in µg m$^{-3}$) relative
to the concentration of BVOC consumed ($\Delta BVOC$ in µg m$^{-3}$), i.e., $Y_{SOA} = \Delta M/\Delta BVOC$. $\Delta M$ was
derived from individual SOA growth curves as shown in Fig. 1. Here the initial mass is defined as
the average SMPS-derived particle mass in the chamber prior to $N_2O_5$ injection, and the final mass
is derived from the maximum of the SOA growth curve when $\Delta BVOC$ stabilizes, as shown in Fig.
S2. Note that under these experimental conditions, SOA formation occurs rapidly, limited on the
short end by the thermal decomposition e-folding lifetime of $N_2O_5$ (~30 s at 295 K) and the e-
folding lifetime of $NO_3$ reaction with γ-terpinene (few milliseconds assuming a rate constant of
$2.9 \times 10^{-11}$ cm$^3$ molecule$^{-1}$ s$^{-1}$), and on the long end by the time scale for heterogeneous uptake of
$N_2O_5$ of several hours assuming an uptake coefficient at low relative humidity of $10^{-4}$ (Abbatt et
al., 2012).
$Y_{SOA}$ with and without seed particles as a function of particle mass loading are depicted in
Fig. 2. The curve shows that under low mass loadings, the yields are less than under high mass
loadings, indicative of absorptive partitioning (Hao et al., 2011;Odum et al., 1996). To model the
measured $Y_{SOA}$ as a function of particle mass loading, we apply an absorptive partitioning model
following the method of Odum et al. (1996), as shown in Eq. (1).
$$Y_{SOA} = M_0 \sum_i \left( \frac{\alpha_i K_{om,i}}{1 + K_{om,i} M_0} \right)$$

Here, $\alpha_i$ is a proportionality constant describing the fraction of product $i$ in the aerosol phase, $M_0$
is the aerosol mass concentration, and $K_{om,i}$ is the absorptive partitioning coefficient of the
absorbing material. Assuming a two-product model, the best fit values are $\alpha_1 = 0.94$, $K_{om,1} =$
$7.9 \times 10^{-4}$, $\alpha_2 = 0.33$, and $K_{om,1} = 2.6 \times 10^{-2}$. Extending this model to a conservative ambient mass
loading of 10 µg m$^{-3}$, characteristic of biogenic SOA-impacted environments (Fry et al., 2014), the
SOA yield is ~10%. We caution that the model is not very well constrained at low mass loadings
due to the limited number of data points below 30 µg m$^{-3}$. From the 95% confidence intervals, a
conservative estimate of the relative uncertainty in the yield at 10 µg m$^{-3}$ is +100/-50%. In contrast,
at mass loadings >500 µg m$^{-3}$, which is more relevant in highly polluted urban areas such as those
along the coast of India (Bindu et al., 2016), $Y_{SOA}$ can be as large as ~50%. For comparison, $Y_{SOA}$
of other reaction systems applying the absorptive partitioning values derived from those
experiments are plotted along with our experimental data in Fig. 2. The γ-terpinene + NO$_3$ $Y_{SOA}$
are significantly less than those involving β-pinene, an important contributor to SOA formation
predominately in the southeastern U.S (Boyd et al., 2015). However, at relatively low particle mass
loadings, $Y_{SOA}$ for NO$_3$ + γ-terpinene is comparable to those derived from the OH oxidation of γ-
terpinene and α-pinene (Griffin et al., 1999;Lee et al., 2006). Interestingly, our measured $Y_{SOA}$ at
comparable mass loadings are also within the reported range of $Y_{SOA}$ from the NO$_3$ oxidation of α-
pinene of 0-16% (see Fry et al. (2014) and references therein), which are relatively small compared
to other monoterpene + NO$_3$ reaction systems, which range from 13% to 65% for β-pinene,
limonene, and Δ-3-carene (Ng et al., 2017). The studies reporting low $Y_{SOA}$ also report relatively
low ON yields and high ketone yields, suggesting that the NO$_3$ oxidation products of α-pinene,
and likely γ-terpinene, lose the nitrate moiety and hence are sufficiently volatile and do not
contribute significantly to SOA formation under atmospherically-relevant aerosol mass loadings.
In contrast, the experiments reporting higher $Y_{SOA}$ report relatively greater ON/ketone yield ratios,
with the exception of sesquiterpenes such as β-caryophyllene, suggesting ON are important aerosol
precursors.

*3.2. Organic nitrate yields*
ON can partition to the particle phase and contribute to SOA formation and mass growth.
However, measurements of their yields are limited and highly variable depending on the
composition of the reactive organic species and the type of oxidant (Ziemann and Atkinson, 2012).
Here we report the measured gas- and aerosol-phase ON yields, and the total (sum of gas and
aerosol ON) yield following γ-terpinene oxidation by $NO_3$. The ON yields ($Y_{ON}$) are defined as the
concentration of ON produced (ΔON) either in the gas or particle phases, relative to the
concentration of BVOC consumed, ΔBVOC, i.e., $Y_{ON}$ = ΔON/ΔBVOC. In these experiments,
ΔBVOC was varied systematically by altering the concentration of $N_2O_5$ added to the chamber
and monitoring the change in BVOC concentration with the GC-FID. These experiments were
conducted both in the presence and absence of $(NH_4)_2SO_4$ seed aerosol particles and under dry
conditions, and corrected for wall losses and dilution.

*3.2.1.  Total gas-phase organic nitrate yield*
As indicated in Fig. 3, the concentration of total gas-phase ON ($ON_g$; determined via FT-
IR) increases linearly as a function of ΔBVOC, independent of the presence or absence of the seed
aerosol. By fitting both the unseeded and seeded data using linear regression, we derive a gas-
phase molar ON yield ($Y_{ONg}$) of 11(±1)%, where the relative uncertainty in the yield of ~9% is
derived from the 95% confidence intervals (shown in dashed lines in Fig. 3) of the linear fit to the
data, and accounting for the measurement uncertainties, shown as error bars. The similar yields
with or without seed particles implies that after some uptake, the two cases might appear identical
to the adsorbing molecules. Some of the variability in the yield presented in Fig. 3, particularly
below $\Delta$BVOC ~ 300 ppb, may be attributed to greater relative uncertainty in $\Delta$ON and $\Delta$BVOC
for low extents of BVOC reaction, different concentrations of $NO_2$ in the chamber, and differences
in the time frame of the experiment, as indicated in Table 1. While some wall loss of the lower
volatility multifunctional oxidation products could bias the reported yields low (Zhang et al.,
2014), the effects of wall loss on the yield of ON are accounted for in these experiments and
minimal (< 5% correction to the yield), given our relatively short experimental timescales (~40
min on average) and measured wall loss rate of the hydroxy nitrate of ~$10^{-5}$ s$^{-1}$. As noted in the
methods section, secondary oxidation of the remaining double bond of $\gamma$-terpinene may account
for ~10% of the uncertainty in $Y_{ONg}$. Regardless, $Y_{ONg}$ observed here for $\gamma$-terpinene is considerably
smaller than those measured from the $NO_3$ oxidation of limonene and $\beta$-pinene, but very similar
to the yield from $NO_3$ oxidation of $\alpha$-pinene (Fry et al., 2014).

*3.2.2.  Total particle-phase organic nitrate yield*

In general, particle-phase ON concentrations ($ON_p$) increase with increasing $\Delta$BVOC as

shown in Fig. 4, with a particle-phase ON yield ($Y_{ONp}$) from the slope of 3($\pm$1)%. Since there were
no significant differences in $ON_p$ between experiments conducted with and without seed aerosol,
the slope (i.e., yield) is derived from a fit to both datasets. The insignificant difference in the
particle phase ON yields between the seeded and unseeded experiments may be due to the large
fraction of organic material in the particles in both cases, and for the seeded experiments, relative
to sulfate. During both the seeded and unseeded experiments, on average particle mass increased
by orders of magnitude following uptake of the oxidation products. Thus, in terms of uptake from
the gas phase, and component solubility, for example, the particles in the two cases are effectively
identical. $Y_{ONp}$ can be affected by wall loss of both semi-volatile ON products and particles. Given
our relatively short experimental timescales and relatively large particle/wall surface area ratios
(upwards of 0.05) compared to other studies (Nah et al., 2016;Zhang et al., 2014), wall loss
corrections amount to an increase in the relative uncertainty of the yield of 8% to 39%. The greater
spread in $ON_p$ compared to $ON_g$ (see Fig. 3) as a function of $\Delta$BVOC may be due to variable
chemistry occurring in the particle phase and the greater relative uncertainty in the case of the
lower particle phase yields. It is possible that the presence of some aerosol liquid water and particle
acidity, aided by the presence of hygroscopic $(NH_4)_2SO_4$ and uptake of product $HNO_3$ by the
particles, could result in relatively lower $ON_p$ yields, even at low relative humidity (Rindelaub et
al., 2015). However, while we did not systematically investigate the dependence of yields on
hydrolysis, we did two experiments that reveal that the ONs produced here are less prone to
hydrolysis. Specifically, we found that the gas (10%), particle (1%-6%), and total ON yields (11%-
16%) at a relative humidity of 50% were within the uncertainty of the yields determined under dry
conditions. The expected major ON product shown in the right-hand side of Fig. 5 has a secondary
nitrooxy functional group, which has been shown to be less prone to hydrolysis than tertiary
nitrooxy groups (Darer et al., 2011).  To account for the effects mentioned above, we estimate a
more conservative aerosol organic nitrate yield of 3 (+2/-1)%, based on the upper limit of the data
variability.

*3.3. Organic nitrate aerosol partitioning and effect on SOA yield*
The sum of $ON_g$ + $ON_p$ ($ON_t$) is plotted as a function of ΔBVOC in Fig. 6. Together, they
result in a total molar ON yield, $Y_{ONt}$ = 14(+3/-2)%, accounting for the potential loss of aerosol
phase ON as described previously, comparable to previously measured ON yields from the $NO_3$
oxidation of α-pinene of 10% (Fry et al., 2014) and 14% (Wangberg et al., 1997). From the ratio
$Y_{ONp}/Y_{ONt}$, ~20% of the total ON produced from γ-terpinene + $NO_3$ partitioned to the particle phase,
for these relatively high aerosol mass loading conditions. Assuming an average ON molar mass of
215 g $mol^{-1}$, representing a $C_{10}$-derived hydroxy nitrate (Rindelaub et al., 2015), roughly 14% of
the total aerosol mass is comprised of ON. Gas-to-particle partitioning depends strongly on the
molecule's equilibrium saturation vapor pressure and mass transfer kinetics (Shiraiwa and
Seinfeld, 2012). The addition of nitrooxy and hydroxy groups, for example, can reduce the
equilibrium saturation vapor pressure by several orders of magnitude (Capouet et al., 2008).
Molecules with saturation vapor pressures >$10^{-5}$ atm are almost exclusively in the gas phase,
whereas those below $10^{-13}$ atm are almost exclusively in the condensed phase (Compernolle et al.,
2011). We can estimate the saturation vapor pressure of the ON ($p_i^0$) based on the estimated ON
aerosol mass fraction ($\varepsilon_i^{aero}$=0.14) as given in Eq. (2) (Valorso et al., 2011).

$$\varepsilon_i^{aero} = \frac{1}{1 + \dfrac{M_{aero}\gamma_i p_i^0}{C_{aero}RT}}$$


Here, $M_{aero}$ is the average particle molar mass, $\gamma_i$ is the activity coefficient of molecule "i", and
$C_{aero}$ is the aerosol mass concentration, R is the gas constant, and $T$ is temperature. Assuming
ideality, i.e., $\gamma_i$=1, $C_{aero}$=835 μg $m^{-3}$ (average of ΔM values from experiments listed in table 1), and
$M_{aero}$=215 g $mol^{-1}$, we derive a $p_i^0$ for ON of ~6×$10^{-7}$ atm or $log_{10}$ saturation concentration of ~4
µg m$^{-3}$, which for a semivolatile $C_{10}$-derived hydrocarbon is expected to have between two and
four oxygen atoms (Donahue et al., 2011). This estimated $p_i^0$ for ON is about an order of magnitude
greater than that calculated for the expected tertiary hydroxy and hydroperoxy nitrates of γ-
terpinene shown in Fig. 5 of $6.9 \times 10^{-8}$ atm and $3.9 \times 10^{-8}$ atm, respectively, using SIMPOL.1
(Pankow and Asher, 2008), suggesting that the $ON_p$ products of γ-terpinene likely comprise a
mixture of hydroperoxy and hydroxy nitrates, and other more volatile ON species, likely keto–
nitrates, e.g. as shown in Fig. 5 for the case of $NO_3$ addition to the more-substituted carbon. For
the keto–nitrate shown in Fig. 5, we calculate a $p_i^0$ value of $1.4 \times 10^{-6}$ atm, using SIMPOL, roughly
a factor of two greater than our estimate for the average for our aerosol. For comparison, the keto–
aldehyde presented in Fig. 5 (γ-terpinaldehyde) has a $p_i^0$ value of 0.092 atm, using SIMPOL. As
presented in the supplementary information, analysis of liquid extracts from filter samples using
UPLC-ESI-ToF-MS/MS operated in negative ion mode indicate the presence of masses consistent
with the first-generation hydroperoxy nitrate and second-generation di-hydroxy di-nitrates in the
aerosol phase, the latter of which may result from both gas- and heterogeneous reactions that
proceed at the unsubstituted olefinic C of a γ-terpinene hydroxy nitrate. In the absence of
substantial $HO_2$ in our experiments, the dominant pathway for $RO_2$ is likely to follow either
$RO_2+NO_3$ or $RO_2+RO_2$ (when [VOC]>>$N_2O_5$). However, in ambient nighttime air there may be
substantially more $RO_2+HO_2$ reactions than in our chamber experiments. Isoprene nitrooxy
hydroperoxide, for example, has been identified as the major product from isoprene oxidation by
the nitrate radical in the presence of $HO_2$ (Schwantes et al., 2015), and organic hydroperoxides
have been identified as major SOA products from monoterpene and sesquiterpene ozonolysis
(Reinnig et al., 2009;Docherty et al., 2005). Thus our chamber experiments may underestimate the
concentration of hydroperoxides formed from γ-terpinene oxidation by $NO_3$ in the ambient
environment. While we did not confirm the presence of epoxides in our experiments, and it is hard
to see how an epoxide could form in the dark gas phase in these experiments, the remaining double
bond of the first-generation hydroxy nitrate may be susceptible to epoxidation in the particle phase.
For example, it is known that peroxyacetyl nitrate (PAN) very efficiently epoxidizes olefins in
solution (Darnall and Pitts, 1970). While there would not be PAN as a product in our experiment,
there could be very significant yields of the corresponding peroxy acyl nitrate from $NO_3$ reaction
with terpinaldehyde, followed by uptake of that compound into the aerosol phase. As shown in
Fig. 7 (A), that PAN compound could then react with, and produce the corresponding epoxide of
any particle-phase compound with a double bond, e.g. the hydroxy nitrate, to produce a $C_{10}H_{17}O_5$
product.  That epoxide would then do a pH-dependent hydrolysis in solution to produce the
corresponding diol ($C_{10}H_{18}O_6$) (Jacobs et al., 2014). Applying the Extended Aerosol Inorganics
Model (E-AIM) (http://www.aim.env.uea.ac.uk/aim/aim.php), we estimate a pH~5.5 for the
$(NH_4)_2SO_4$ seed particles under saturated conditions, but becoming more acidic as the particles
uptake $HNO_3$.  It is important to note that the reaction products and their concentrations and thus
degree of aerosol partitioning and SOA yields may also be affected by the concentration of $NO_3$
in the chamber. Under very high $NO_x$ conditions as in some of the experiments here, reactions
between $RO_2$ and $NO_3$ out-compete those with $HO_2$, which may lead to formation of relatively
more volatile carbonyl reaction products, as indicated in Fig. 5, and relative suppression of particle
mass. This effect is consistent with other studies that report lower SOA yields in the presence of
high $NO_x$ concentrations (Ng et al., 2007;Presto et al., 2005;Song et al., 2005). Regardless, an
aerosol mass fraction of ON of 14% is considerably less than that obtained for other monoterpenes
reacting with $NO_3$, with the exception of α-pinene (Fry et al., 2014). This could be a result of both
production of mostly volatile ON species, in particular keto–nitrates, and further reaction of the
olefinic hydroxy nitrate in the aerosol phase. To verify the potential role of hydroxy nitrates in
SOA production from $NO_3 + \gamma$-terpinene as well as the presence of other ON, the following section
focuses on product identification of gas phase ON species using CIMS and determination of gas
phase hydroxy nitrate yields.

*3.4. CIMS product identification and hydroxy nitrate yields*
$NO_3$ reactions with VOCs lead to either abstraction of a hydrogen atom or addition to a double
bond. Since $\gamma$-terpinene has two double bonds with similar character, $NO_3$ likely has equal
probability of adding to either internal double bond. However, addition of $NO_3$ to either one of the
olefins is likely to form the more stable tertiary nitrooxy alkyl radical. Subsequent addition of $O_2$
forms the $\beta$-nitrooxyperoxy radical that can lead to an array of products, including hydroxy
nitrates, most likely from self or cross $RO_2 + RO_2$ reactions or isomerization (Yeh and Ziemann,
2014;Ziemann and Atkinson, 2012). $C_{10}$-derived hydroxy nitrates and other multifunctional ON
have been identified in field-sampled SOA particles, and for nighttime $ON_p$, $C_{10}$-derived ON could
account for approximately 10% of the organic aerosol mass during the Southern Oxidant and
Aerosol Study (SOAS) campaign in the U.S. southeast (Xu et al., 2015;Lee et al., 2016). However,
our current understanding of $C_{10}$-derived hydroxy nitrate yields is limited to production via
oxidation of $\alpha$-pinene (Wangberg et al., 1997). Here we expand on this by determining the hydroxy
nitrate yield from $\gamma$-terpinene oxidation by $NO_3$ and identify other potentially important ON
species using CIMS.
Figure 8 shows example CIMS mass spectra (red and blue traces) and the enhancements
over the background in the presence of $NO_3$ (black trace) following a chamber experiment, where
the enhancement is calculated from the signal for $\frac{NO_3 - no\ NO_3}{no\ NO_3}$. Several molecules were detected at
masses below the iodide reagent ion signal ($m/z = 127$) following $N_2O_5$ addition to the chamber
and correspond to $NO_3^-$ ($m/z = 62$), $NO_3^-\cdot(H_2O)_{1,2}$ ($m/z = 80, 98$), $N_2O_5^-$ ($m/z = 108$), and what
appears to be a nitrate–nitric acid cluster anion, $HN_2O_6^-$ ($m/z = 125$) (Dubowsky et al., 2015; Huey,
2007). The water cluster ions and nitric acid (also at $m/z = 190$, corresponding to $I^-\cdot HNO_3$) result
from ion–molecule reactions in the humidified drift tube of our CIMS and residual $HNO_3$ from the
$N_2O_5$ cold trap. Several larger molecular weight species were detected in the range of $300 \le m/z \le$
450, consistent with products from monoterpene oxidation, with enhancements over the
background as large as a factor of 50 to 100. Specifically, the first-generation hydroxy nitrates are
observed at $m/z = 342$ ($C_{10}H_{17}NO_4–I^-$). Several masses follow, separated by 16 mass units, or
addition of a single oxygen atom, whereby each new ON has 15, 17, or 19 H atoms. Similar
observations were made in the field during the SOAS campaign for both $ON_g$ and $ON_p$ (Lee et al.,
2016), indicating the presence of highly-functionalized ONs. It is important to note that the
products observed here are derived from a single monoterpene, whereas the field ON
measurements consist of products derived from all ambient monoterpene oxidation. Other major
peaks included those at $m/z = 340$, potentially representing an iodide-adduct with either an
aldehyde or keto–nitrate ($C_{10}H_{15}NO_4–I^-$), and $m/z = 358$, which may be indicative of an iodide-
adduct with a hydroperoxy nitrate ($C_{10}H_{17}NO_5–I^-$). A cluster of ions was detected above $m/z = 400$,
potentially representing molecules with higher degrees of oxygenation and secondary oxidation
products such as a di-hydroxy–di-nitrate at $m/z = 421$ ($C_{10}H_{18}N_2O_8–I^-$), which could be formed
through second-generation oxidation at the remaining unsubstituted carbon of the double bond on
the first-generation hydroxy nitrate. It is important to note that the CIMS sensitivity for each of
these species is likely different and depends on the polarity and acidity of the individual compound,
which is affected by the type and positions of the different functional groups (Lee et al., 2016).
For example, iodide-adduct CIMS is not particularly sensitive to aldehyde and carbonyl nitrates,
whereas more acidic and polar molecules such as hydroxy nitrates and carboxylic acids can exhibit
much greater sensitivity (Lee et al., 2016). Moreover, in general as the molecular size and number
of oxygenated groups increase (particularly –OH groups), the sensitivity also increases. Hence,
without commercial or custom synthetic standards, no quantitative analysis of the array of ON
products could be reliably performed using this technique.

Here we determine the yield of γ-terpinene-derived hydroxy nitrates. Since there is no

commercially-available standard for the expected first-generation γ-terpinene hydroxy nitrate, we
use a synthetic olefinic hydroxy nitrate derived from α-pinene (structure shown in Fig. S4) for
quantitative analysis (Rindelaub et al., 2016). It is possible that the CIMS is less sensitive to this
nitrate compared to the more sensitive α,β-hydroxy nitrate structure expected of the first-
generation γ-terpinene hydroxy nitrates, similar to the differences in the CIMS sensitivity for 4,3-
isoprene hydroxy nitrate (4,3-IN) and 1,4-IN (Xiong et al., 2015). However, the use of an olefinic
hydroxy nitrate is consistent with that expected from γ-terpinene oxidation because of its diolefinic
character. As shown in Fig. 9, γ-terpinene-derived hydroxy nitrate concentrations increase linearly
over the range of ΔBVOC with a hydroxy nitrate yield defined from the slope as 4(±1)%.
Assuming the CIMS sensitivity for the γ-terpinene hydroxy nitrates may be a factor of three greater
than for our synthetic α-pinene-derived hydroxy nitrate, a more conservative estimate of the γ-
terpinene-derived hydroxy nitrate yield is 4(+1/-3)%. To our knowledge, the only monoterpene
hydroxy nitrate yield to have been quantified following $NO_3$ oxidation is 2-hydroxypinan-3-
nitrate, derived from α-pinene (Wangberg et al., 1997). In that study, the hydroxy nitrate yield was
determined using a combination of FT-IR and GC-ECD to be 5(±0.4)%, on the same order as the
yield presented in this study for γ-terpinene using CIMS. 3-oxopinan-2-nitrate ($C_{10}H_{15}NO_4$; 213 g
mol$^{-1}$) and a short-lived, thermally unstable peroxy nitrate ($C_{10}H_{16}N_2O_7$; 276 g mol$^{-1}$) were also
identified in that study. It is possible that similar products are made following NO$_3$ oxidation of γ-
terpinene, and potentially make up the signals detected at *m/z* = 340 and *m/z* = 403, respectively,
as shown in Fig. 8. However, the CIMS sensitivity toward these products is expected to be
relatively small compared to that for the hydroxy nitrates, due to their relatively lower polarity and
acidity. Moreover, peroxy nitrates are thermally unstable and their concentrations are likely greatly
reduced during transfer through the heated sampling line.

*3.5.Proposed reaction mechanism*
The similarities between the, at first seemingly low, γ-terpinene + NO$_3$-derived $Y_{ONt}$, hydroxy
nitrate yield, and $Y_{SOA}$ with those for NO$_3$ + α-pinene are provocative. This suggests the two
monoterpenes may undergo very similar degradation pathways following NO$_3$ oxidation, which is
not observed with other monoterpenes with a substituted endocyclic double bond (Fry et al., 2014).
As such, our mechanistic interpretation, shown in Fig. 5, is analogous to that for the α-pinene +
NO$_3$ reaction, as described in the Master Chemical Mechanism (MCM) (Jenkin et al.,
1997;Saunders et al., 2003). NO$_3$ will predominately add to the C-3 (unsubstituted) position
forming the more stable tertiary alkyl radical. However, to some extent, NO$_3$ may also add to the
second carbon forming the less stable secondary alkyl radical, approximately 35% of the time
according to the MCM. Oxygen promptly adds to the alkyl radical to form either a tertiary or
secondary peroxy radical (ROO·). Excess NO$_2$, due to thermal decomposition of N$_2$O$_5$, can add to
the peroxy radical forming a thermally unstable peroxy nitrate (-OONO$_2$) in equilibrium with the
peroxy radical. Subsequent RO$_2$· self- and cross-reactions as well as reaction with NO$_3$ form the
alkoxy radical (RO·). The alkoxy radical can subsequently decompose to form a carbonyl nitrate
or γ-terpinaldehyde (Fig. 5) and release $NO_2$. Analogously, pinonaldehyde is the major $NO_3$
oxidation product of α-pinene with reported yields of 62(±4)% (Wangberg et al., 1997) and 75±6%
(Berndt and Böge, 1997). Given the very similar tertiary alkoxy radicals produced from $NO_3$
oxidation of α-pinene and γ-terpinene, and the similar SOA, $ON_t$, and hydroxy nitrate yields,
conceivably γ-terpinaldehyde is produced and with similarly high but undetermined yields as
pinonaldehyde from α-pinene oxidation by $NO_3$. Similar results have been reported for the
ozonolysis of γ-terpinene, which primarily leads to decomposition and formation of γ-
terpinaldehyde with a yield of 58% (Ng et al., 2006). Alternatively, disproportionation, involving
a secondary peroxy radical, produces a hydroxy nitrate and a carbonyl compound from the
partnering $RO_2$ (Yeh and Ziemann, 2014;Ziemann and Atkinson, 2012;Wangberg et al., 1997). As
we have shown, the experimentally-derived yield for these products is 4%, or roughly 25% of $ON_t$.
The remaining organic nitrate species likely contains both carbonyl and hydroperoxy (-OOH)
functionalities, and perhaps peroxy nitrates, following $NO_2$ addition to the peroxy radical. A major
species detected by our CIMS has an $m/z = 358$, which may represent an $I^-$ adduct with a
hydroperoxy nitrate. This product is only produced due to reactions between hydroperoxy radicals
($HO_2\cdot$) and $RO_2\cdot$ (Ziemann and Atkinson, 2012). Conceivably, $HO_2\cdot$ is produced in our system
from hydrogen abstraction from alkoxy radicals by oxygen (Wangberg et al., 1997).

To test the hypothesis that γ-terpinene behaves similarly to α-pinene following reaction

with $NO_3$, we ran a simple box model based on the mechanisms for $NO_3$ oxidation of α-pinene as
presented in the MCM, and compared the model output with the measured concentrations of γ-
terpinene, $NO_2$, and hydroxy nitrates. The model is constrained by the initial and final GC-FID-
derived concentrations of γ-terpinene. Since the nitrate radical concentration was not determined
experimentally, the concentration of $NO_3$ in the model was determined by adjusting the $N_2O_5$
concentration until the fitted concentration change of γ-terpinene matched that which was
measured. This approach implicitly assumes γ-terpinene is consumed only from reaction with $NO_3$,
which is expected given the orders of magnitude greater reactivity of $NO_3$ compared to the other
reactants in our system, which includes $N_2O_5$ and $NO_2$. The results of the model are presented in
Fig. 10. For comparison, modeled concentrations are plotted along with the measured
concentrations of γ-terpinene, $NO_2$, and hydroxy nitrates derived from one of the experiments. At
a first approximation, the modeled concentrations appear to be in agreement with those measured,
given the semi-quantitative nature of the product, particularly the hydroperoxides. As shown in the
top panel of Fig. 10, $NO_3/HO_2$ ratios are ~3 at peak [$HO_2$], then decrease to ~1 as the products
reach steady state.  In comparison, ambient nighttime $NO_3/HO_2$ ratios of ~1 have been measured
during the PROPHET 1998 field intensive in northern Michigan (Hurst et al., 2001;Tan et al.,
2001), and ~0.25 at the BEARPEX field site in north central California (Bouvier-Brown et al.,
2009;Mao et al., 2012). The relatively larger ratios in our chamber, initially, suggest hydroperoxy
nitrates may be underrepresented compared to the atmosphere. Notably, the agreement between
modeled and measured [$NO_2$] implies that model-derived [$N_2O_5$] is close to that in the reaction
chamber as [$NO_2$] is in equilibrium with $N_2O_5$. Although not quantified experimentally, qualitative
analysis of the CIMS mass spectra indicates the presence of carbonyl and hydroperoxy nitrates,
which is consistent with the major ON products expected from the mechanism shown in Fig. 5.

**4.  Atmospheric Implications**
The relatively low SOA and ON yields observed here under dry conditions at ambient mass
loadings suggests γ-terpinene may not be an important SOA precursor at night, when $NO_3$ can be
the dominant oxidant. However, the low saturation vapor pressure of the hydroxy nitrates, which
constitute a significant portion of the total ON, and the presence of some highly oxygenated
products, further suggests that these molecules are potentially important contributors to SOA mass.
While our experiments were conducted near dry conditions, in the ambient forest environment,
particularly at night and in the early morning when the relative humidity near the surface is high
and $NO_3$ reactions are competitive with $O_3$ and OH, hydroxy nitrates in the particle phase can
enhance SOA formation through acid-catalyzed hydrolysis and oligomerization, and in the
presence of sulfates, form organic sulfates (Liu et al., 2012;Paulot et al., 2009;Rindelaub et al.,
2016;Surratt et al., 2008;Rindelaub et al., 2015), ultimately affecting the lifetime of $NO_x$ (Browne
and Cohen, 2012;Xiong et al., 2015). Furthermore, the transformation of the nitrooxy group to a
hydroxyl or sulfate group will alter the hygroscopicity of the particle, making them more effective
cloud condensation nuclei (Suda et al., 2014).
It is important to note that under relatively clean air conditions, the peroxy radical produced
via $NO_3$ reaction with γ-terpinene will often react with $HO_2$ to produce nitrooxy hydroperoxides.
As shown in the reaction scheme (B) in Fig. 7, these species can then react with $NO_3$ and then
$HO_2$, $RO_2$ or $NO_3$ again, to yield a variety of highly oxidized very low vapor pressure products
that will likely partition completely to the aerosol phase. Under humid conditions, the nitrooxy
groups may hydrolyze, leaving more polar/water soluble OH groups.
Although the SOA yields are low, these chamber experiments did not represent all possible
reactants that can produce particle phase precursors. Recent work indicates keto–aldehydes are
potentially an important source of nitrogen-containing low volatility compounds following their
reaction with dimethylamine, serving as precursors to SOA and brown carbon (Duporté et al.,
2016). As shown in this study, the keto–aldehyde yield is expected to be large, along with other
internal olefinic terpenes. It is also important to note that the keto–aldehyde product, γ-
terpinaldehyde, is olefinic. Further homogeneous and multiphase oxidation reactions at the
remaining reactive double bond can potentially transform these species into oligomeric lower-
volatility oxidation products, adding to the overall SOA burden (Liggio and Li, 2008). In regions
such as the northern U.S., where there are greater proportions of polyolefinic monoterpenes (Geron
et al., 2000), γ-terpinene may be an important reactive VOC, and thus impact aerosol and local-
scale $NO_x$.

**5.  Conclusions**
The total molar ON yield from the $NO_3$ oxidation of γ-terpinene was found to be 14(+3/-2)%.
Relatively low particle-phase ON and SOA yields are consistent with previous studies that show
SOA yields are generally dependent on the yield of ON. Although γ-terpinene is a diolefin, the
ON, hydroxy nitrate, and SOA yields are similar to those for α-pinene oxidation by $NO_3$.
Considering the position of the two double bonds, the expected major product is γ-terpinaldehyde,
which is considerably more volatile than the ON products. Box model calculations that assume
large keto–aldehyde yields are also in agreement with the measured concentrations of hydroxy
nitrates, suggesting very similar mechanistic behavior to that of α-pinene oxidation. Several gas-
and particle-phase ON products have been inferred from mass spectrometry analysis, indicating
that $NO_3$ reaction with γ-terpinene may be an important source of ON and dicarbonyl compounds
in forest-impacted environments.

**Author Contributions**
J. H. S. and P. B. S. designed the research and wrote the manuscript. J. H. S. performed the yield
experiments and analyzed the data. J. H. S. and C. d-P. analyzed the filter samples. L. L. oversaw
the analysis of the filter samples. All authors contributed intellectually to the manuscript.

**Acknowledgements**
J. H. Slade and P. B. Shepson acknowledge support from the National Science Foundation grant
CHE-1550398. The authors declare that they have no conflicts of interests.
**Tables**
Table 1. Initial conditions and yields from individual experiments. Time indicates the period
between $N_2O_5$ addition to the chamber and gas and particle collection by the denuder and filter.
"n.m." indicates "not measured".

| Date | Seed | $\Delta BVOC/$ ppb | $\Delta ON_g/$ ppb | $\Delta BVOC/$ mol×10⁻⁴ | $\Delta ON_p/$ mol×10⁻⁴ | $[NO_2]/$ ppb | Time/ min | $Y_{ONg}$ | $Y_{ONp}$ | $\Delta M/$ µg m⁻³ |
|---|---|---|---|---|---|---|---|---|---|---|
| 9/9/15 | None | 229 | 10 | 0.52 | 0.012 | 60 | 52 | 4% | 2% | 530 |
| 9/17/15 | None | 131 | 16 | 0.30 | 0.025 | 31 | 30 | 12% | 9% | 272 |
| 9/19/15 | None | 90 | 7 | 0.20 | 0.017 | 29 | 28 | 7% | 8% | 311 |
| 9/21/15 | None | 214 | 15 | 0.48 | 0.007 | 56 | 73 | 7% | 2% | 604 |
| 9/23/15 | None | 256 | 21 | 0.58 | 0.017 | 82 | 18 | 8% | 3% | 534 |
| 9/23/15 | None | 80 | 8 | 0.18 | 0.007 | 23 | 31 | 10% | 4% | 61 |
| 10/20/15 | $(NH_4)_2SO_4$ | 761 | 90 | 1.70 | 0.038 | n.m. | 62 | 12% | 2% | 3800 |
| 10/22/15 | $(NH_4)_2SO_4$ | 164 | 31 | 0.37 | 0.035 | n.m. | 48 | 19% | 9% | 575 |
| 10/28/15 | $(NH_4)_2SO_4$ | 47 | 9 | 0.11 | 0.002 | 7 | 14 | 18% | 2% | 32 |
| 11/09/15 | $(NH_4)_2SO_4$ | 245 | 23 | 0.55 | 0.006 | 54 | 48 | 10% | 1% | 1143 |
| 11/10/15 | $(NH_4)_2SO_4$ | 66 | 4.5 | 0.15 | 0.006 | 28 | 32 | 7% | 4% | 159 |
| 11/12/15 | None | 413 | 49 | 0.93 | 0.020 | 326 | 45 | 12% | 2% | 623 |
| 11/18/15 | None | 408 | 39 | 0.92 | 0.037 | 138 | 35 | 10% | 4% | 2206 |


**Figures**

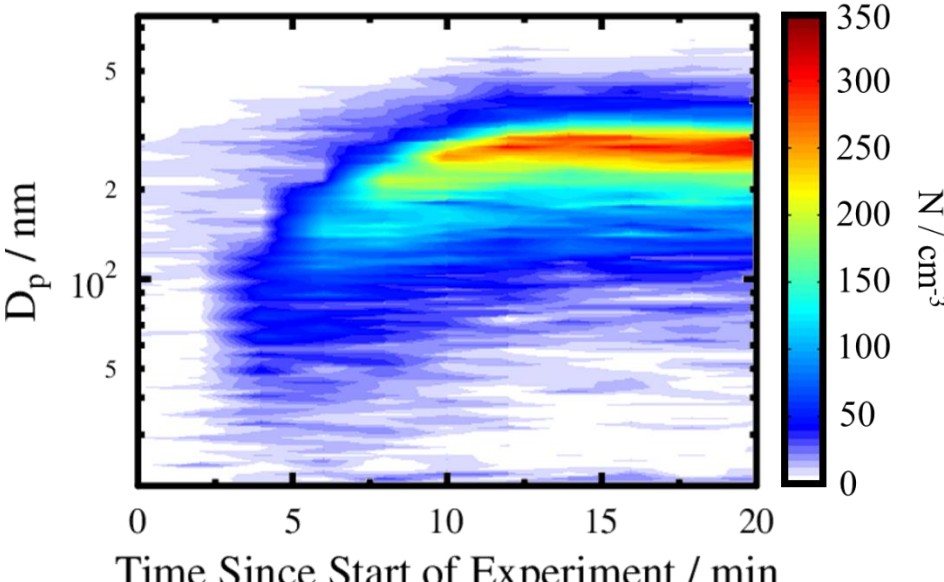


Figure 1. Example wall loss-corrected SOA growth curve for γ-terpinene + $NO_3$ in the absence of
seed aerosol. The color scale represents aerosol number concentration, N (cm⁻³).

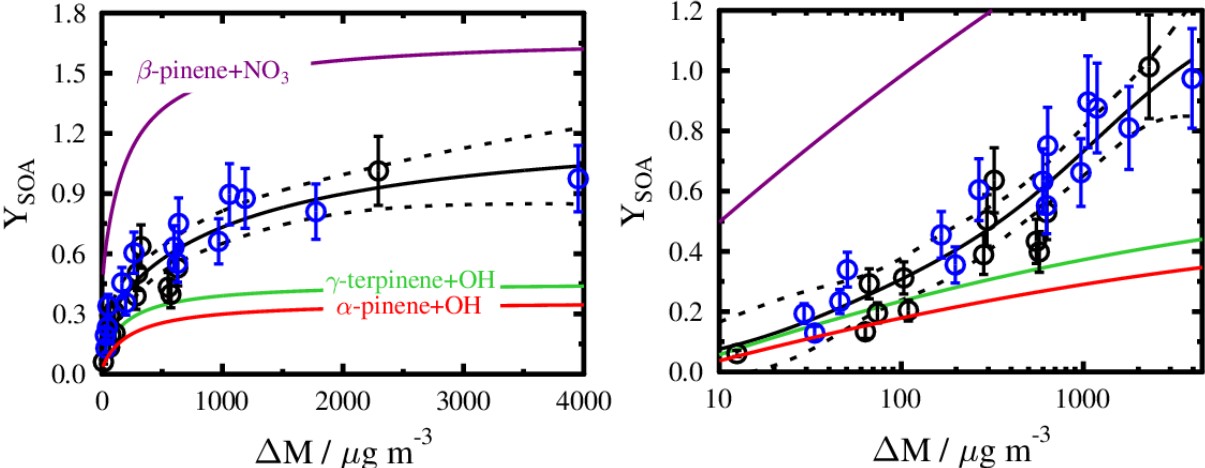


Figure 2. Change in aerosol mass concentration ($\Delta M$) and wall-loss corrected SOA yields ($Y_{SOA}$)

from the NO$_3$ oxidation of γ-terpinene in unseeded (black circles) and (NH$_4$)$_2$SO$_4$-seeded

experiments (blue circles). The data were fitted to a two-product absorptive partitioning model

(black curve) and the dashed curves represent the 95% confidence intervals of the fitting function.

For comparison, the mass-dependent yield curves of α-pinene and γ-terpinene in the presence of

OH are shown in the red and green curves, respectively, and β-pinene + NO$_3$ in purple (Griffin et

al., 1999;Lee et al., 2006). For clarity, the right panel shows the left panel data on a log scale.

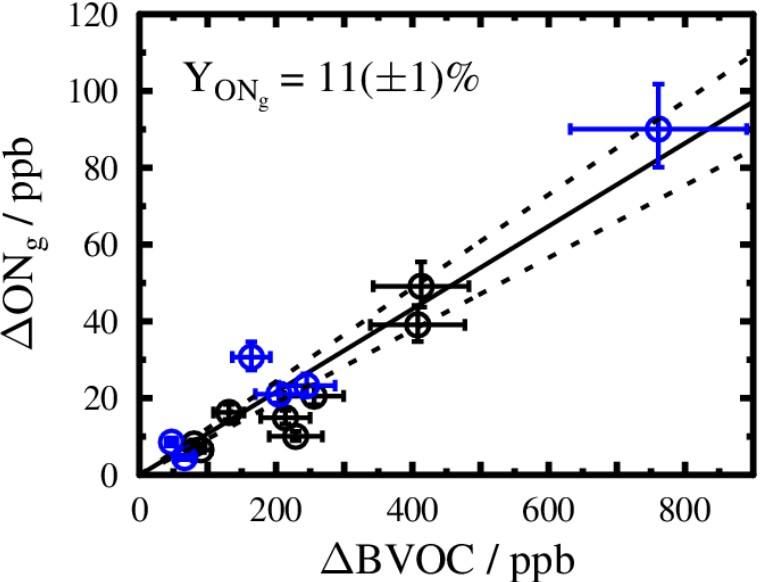


Figure 3. Total wall loss- and dilution-corrected gas-phase organic nitrate production ($\Delta ON_g$) as a
function of the amount of BVOC consumed ($\Delta BVOC$) for the unseeded (black circles) and
$(NH_4)_2SO_4$-seeded experiments (blue circles). Horizontal and vertical error bars represent the
uncertainty in the GC-FID and FT-IR calibrations, respectively. The black line shows the linear fit
of the data through the origin and the dashed lines indicate the 95% confidence intervals of the fit.
The slopes of these lines represent the fractional organic nitrate yield and uncertainty presented in
the plot, respectively.

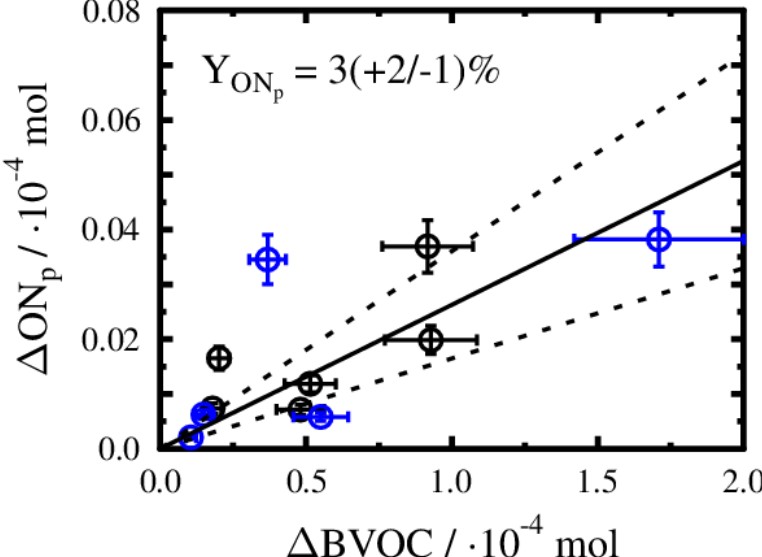


Figure 4. Total wall loss- and dilution-corrected particle-phase organic nitrate production ($\Delta ON_p$)
as a function of the amount of BVOC consumed ($\Delta BVOC$) for the unseeded (black circles) and
$(NH_4)_2SO_4$-seeded experiments (blue circles). The error bars and fits are derived as in Fig. 3.


Figure 5. Proposed initial reaction pathways for the NO₃ oxidation of γ-terpinene. For simplicity,

only the first-generation oxidation products are shown.

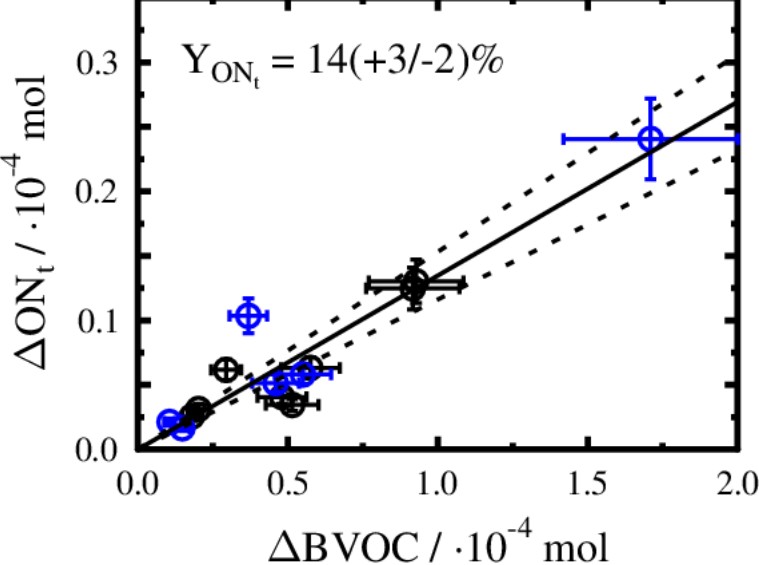


Figure 6. Total wall loss- and dilution-corrected organic nitrate production ($\Delta ON_t$) as a function
of the amount of BVOC consumed ($\Delta BVOC$) for the unseeded (black circles) and $(NH_4)_2SO_4$-
seeded experiments (blue circles). The error bars and fits are derived as in Fig. 3.

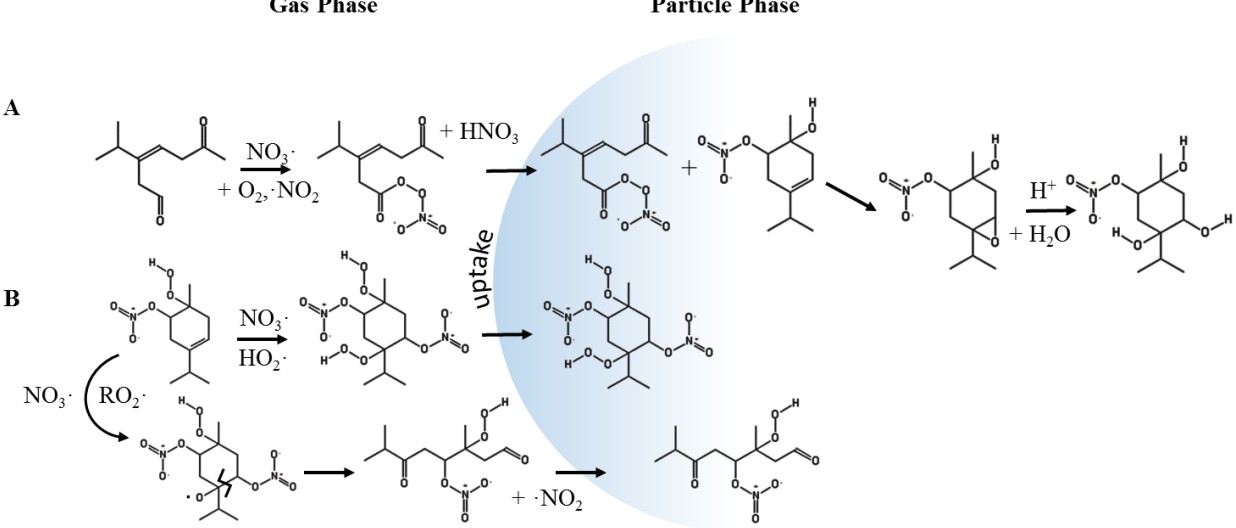


Figure 7. Potential second-generation oxidation reactions and particle-phase chemistry. (A) $NO_3$
oxidation of terpinaldehyde and olefin epoxidation by peroxy acyl nitrate, followed by acid-
catalyzed hydrolysis of the epoxide to form the diol. (B) $RO_2\cdot$ and $HO_2\cdot$ pathways following $NO_3$
oxidation of the first-generation nitrooxy hydroperoxide.


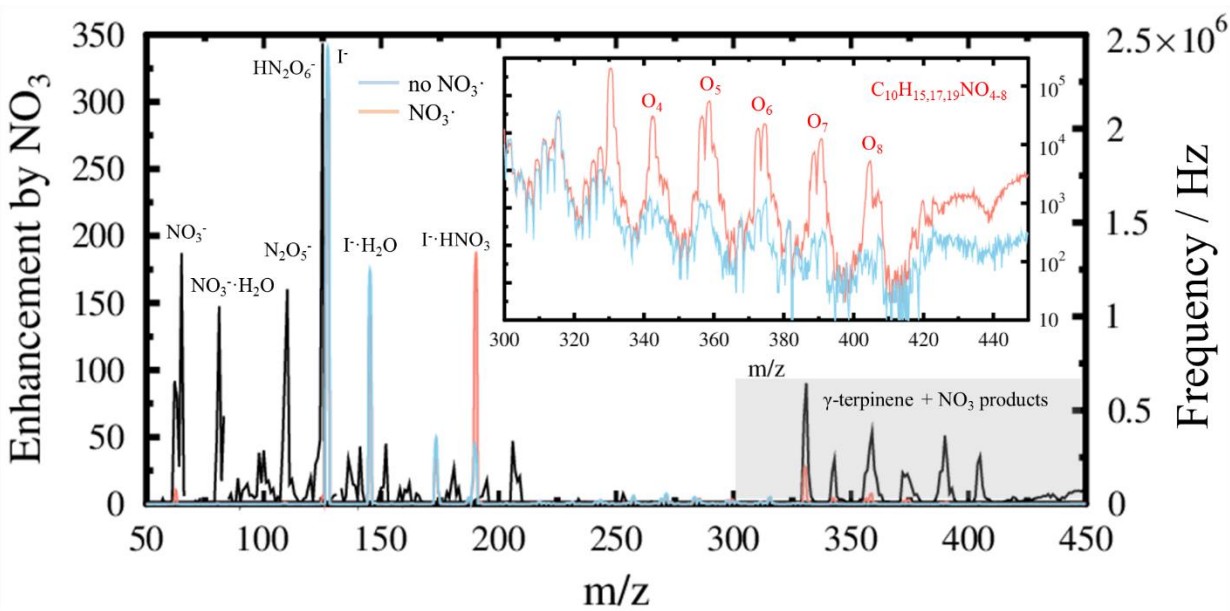


Figure 8. CIMS mass spectra before (blue) and after γ-terpinene oxidation by $NO_3$ (red) correspond
to right axis. Signal enhancement by addition of $NO_3$ is shown in the black trace. The inset figure
shows an enhanced region of the mass spectra corresponding to the shaded area, which indicates
the presence of multifunctional ON compounds with the number of oxygen atoms consistent with
the depicted chemical formula. The "$O_4$" peak was used to quantify hydroxy nitrate concentrations.
Signal enhancement by addition of $NO_3$ is shown in the black trace.

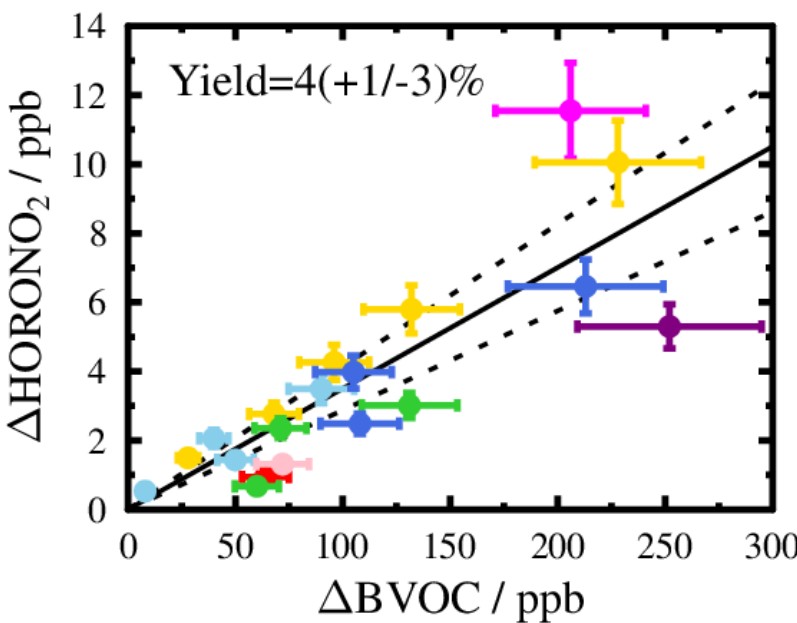


Figure 9. γ-terpinene hydroxy nitrate production (ΔHORONO$_2$) as a function of the amount of
BVOC consumed (ΔBVOC). Colors represent independent experiments performed on different
days. All of the experiments were conducted in the absence of seed aerosol. The error bars and fits
are derived as in Fig. 3.

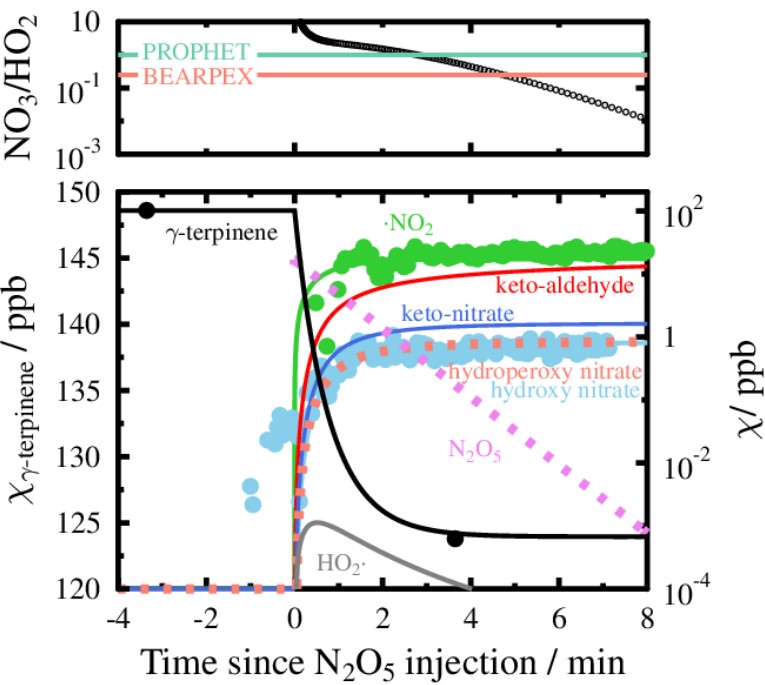

Figure 10. Time series of experiment indicating measured (circles) and modeled (lines) concentrations of γ-terpinene (black), $N_2O_5$ (dashed violet), $NO_2$ (green), $HO_2$ (gray), keto-aldehyde (red), keto-nitrate (dark blue), hydroperoxy nitrate (dashed pink), and hydroxy nitrate (light blue). Top panel shows simulated $NO_3/HO_2$ ratios (black circles) compared to measured ambient nighttime ratios from the PROPHET and BEARPEX field intensives. The model is based on the MCM for α-pinene reaction with $NO_3$.

 **Supplementary Information**

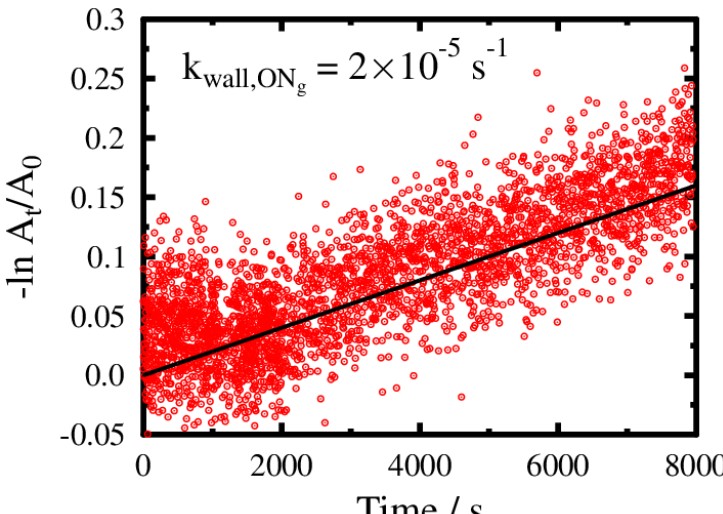


Figure S1. Wall loss rate of m/z 342 as measured by CIMS, corresponding to the monoterpene
hydroxy nitrate–I⁻ adduct.

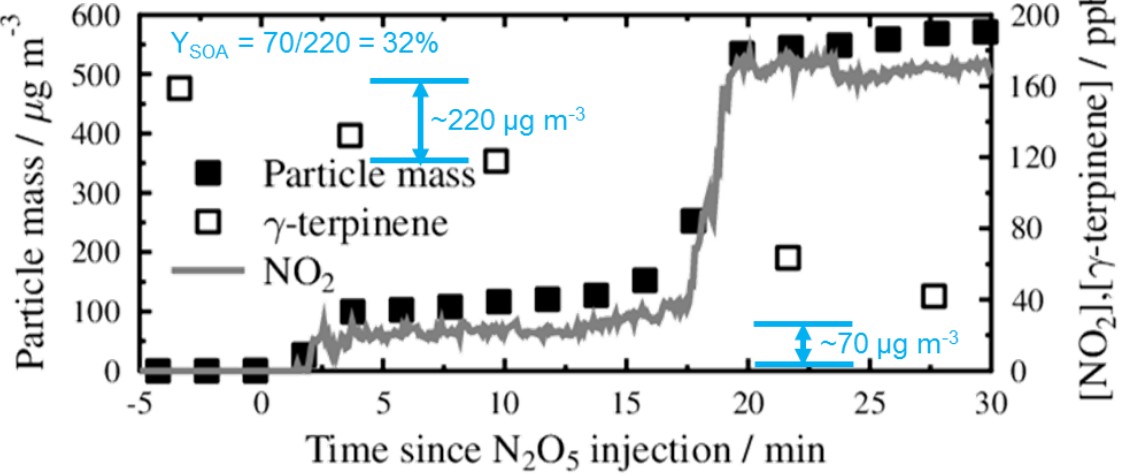


Figure S2. Example experimental time series and calculation of SOA yield.

*Identification of ON in filter extracts*

Figure S3 shows the extracted ion chromatograms (EIC) of the synthetic α-pinene-derived

hydroxy nitrate (red), and potential particle-phase organic nitrates, including the first-generation
hydroperoxy nitrate (green) and di-hydroxy di-nitrate (blue). For each EIC, there is a
corresponding MSMS (MS$^2$) spectrum, which shows the fragment ions of the parent [M+AcO$^-$]
adduct ion species. The synthesized α-pinene-derived hydroxy nitrate adduct with AcO$^-$ ($m/z$ =
274.1291) MS$^2$ spectrum indicates there are two primary fragment ions that correspond to AcO$^-$
and NO$_3^-$. These ions were then used as signatures to identify organic nitrate species in the filter
sample extracts. Detected masses and their corresponding mass spectra and tandem mass spectra
were further analyzed and matched according to the chemical formula: $C_aH_bN_cO_d$. Of the two
samples analyzed, the most abundant species with the NO$_3^-$ fragment ion have an $m/z$ = 353.1197,
corresponding to a molecule with chemical formula $C_{10}H_{18}N_2O_8$ + AcO$^-$ , and $m/z$ = 290.1241
($C_{10}H_{17}NO_5$ + AcO$^-$).

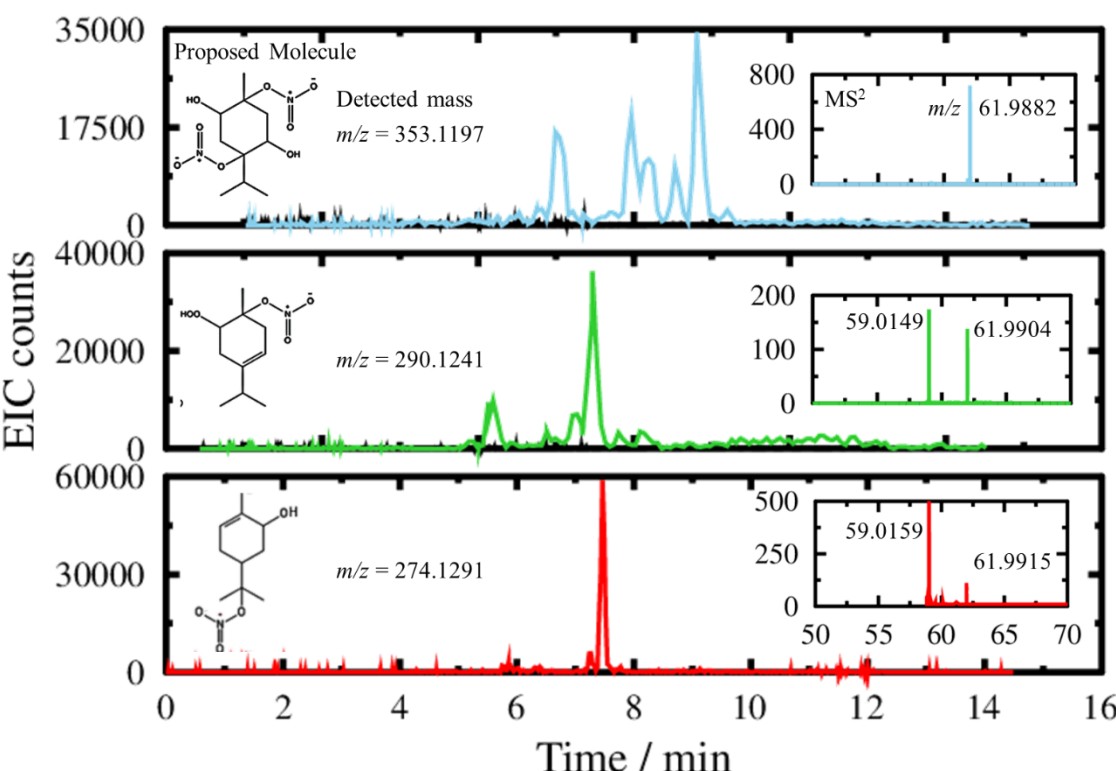


Figure S3. UPLC-ESI(-)-ToF-MS/MS extracted ion chromatograms (EIC) for the synthetic α-
pinene-derived hydroxy nitrate (red), hydroperoxy nitrates present in the filter extracts (green),
and di-hydroxy–di-nitrates present in the filter extracts (blue). For reference, the background EICs
(HPLC-grade methanol) for each mass are also plotted (black). The insets show the MS$^2$ spectra
and the listed $m/z$ values in the MS$^2$ spectra correspond to the most intense peak. Potential
molecular structures are shown for reference.

Figure S4. Molecular structure of the synthetic α-pinene-derived hydroxy nitrate used for
calibration of the CIMS.
*Box model inputs*
The box model applied to simulate the reaction and products from the NO$_3$ oxidation of γ-terpinene
was performed in Matlab v7.7.0 using the ordinary differential equations (ODE23s) solver in
Matlab. Table S1 lists the various reactions and rate constants applied in the model. The majority
of the rate constants were abstracted from those applied in the Master Chemical Mechanism
version, with the exception of NO$_3$ + γ-terpinene since the MCM does not explicitly include γ-
terpinene. Wall loss rate constants were included in the model for NO$_3$, N$_2$O$_5$, and the hydroxy-,
hydroperoxy-, and keto-nitrates as described in the footnotes of Table S1.
**Table S1.** Box model parameters for simulating the NO$_3$+γ-terpinene reaction in the chamber.

| Reaction | Rate constant |
|---|---|
| N$_2$O$_5$ → NO$_3$ + NO$_2$ | $\dfrac{1\times10^{-12}}{2.13\times10^{-27}e^{\frac{11025}{T}}}$ cm$^3$ molecule$^{-1}$ s$^{-1}$ |
| NO$_3$ + NO$_2$ → N$_2$O$_5$ | $1 \times 10^{-12}$ cm$^3$ molecule$^{-1}$ s$^{-1}$ |
| [a]NO$_3$ + wall → loss | $6 \times 10^{-4}$ s$^{-1}$ |
| [b]N$_2$O$_5$ + wall → loss | $5 \times 10^{-6}$ s$^{-1}$ |
| [c]γ-terpinene + NO$_3$ → α-nitrooxy peroxy radical | $24 \times 10^{-12}$ cm$^3$ molecule$^{-1}$ s$^{-1}$ *0.35 |

| | |
|---|---|
| α-nitrooxy peroxy radical + NO$_3$ → α-nitrooxy alkoxy radical + NO$_2$ | $2.3 \times 10^{-12}$ cm$^3$ molecule$^{-1}$ s$^{-1}$ |
| α-nitrooxy peroxy radical + HO$_2$ → β-hydroperoxy nitrate | $2.91 \times 10^{-13} e^{\frac{1300}{T}}$ cm$^3$ molecule$^{-1}$ s$^{-1}$ *0.914 |
| α-nitrooxy peroxy radical + RO$_2$ → β-hydroxy nitrate | $2.5 \times 10^{-13}$ cm$^3$ molecule$^{-1}$ s$^{-1}$ *0.1 |
| α-nitrooxy peroxy radical + RO$_2$ → α-nitrooxy alkoxy radical | $2.5 \times 10^{-13}$ cm$^3$ molecule$^{-1}$ s$^{-1}$ *0.8 |
| α-nitrooxy peroxy radical + RO$_2$ → β-keto nitrate | $2.5 \times 10^{-13}$ cm$^3$ molecule$^{-1}$ s$^{-1}$ *0.1 |
| α-nitrooxy alkoxy radical + O$_2$ → β-keto nitrate + HO$_2$ | $2.5 \times 10^{-14} e^{\frac{-300}{T}}$ cm$^3$ molecule$^{-1}$ s$^{-1}$ |
| α-nitrooxy-β-alkoxy radical → keto-aldehyde + NO$_2$ | $4 \times 10^5$ s$^{-1}$ |
| [c]γ-terpinene + NO$_3$ → β-nitrooxy peroxy radical | $24 \times 10^{-12}$ cm$^3$ molecule$^{-1}$ s$^{-1}$ *0.65 |
| β-nitrooxy peroxy radical + NO$_3$ → β-nitrooxy alkoxy radical + NO$_2$ | $2.3 \times 10^{-12}$ cm$^3$ molecule$^{-1}$ s$^{-1}$ |
| β-nitrooxy peroxy radical + HO$_2$ → α-hydroperoxy nitrate | $2.91 \times 10^{-13} e^{\frac{1300}{T}}$ cm$^3$ molecule$^{-1}$ s$^{-1}$ *0.914 |
| β-nitrooxy peroxy radical + RO$_2$ → α-hydroxy nitrate | $6.7 \times 10^{-15}$ cm$^3$ molecule$^{-1}$ s$^{-1}$ *0.1 |
| β-nitrooxy peroxy radical + RO$_2$ → β-nitrooxy alkoxy radical | $6.7 \times 10^{-15}$ cm$^3$ molecule$^{-1}$ s$^{-1}$ *0.9 |
| β-nitrooxy alkoxy radical → keto-aldehyde + NO$_2$ | $1 \times 10^6$ s$^{-1}$ |
| [d]Wall loss rate of hydroxy nitrate, keto nitrate, and hydroperoxy nitrate | $2 \times 10^{-5}$ s$^{-1}$ |

[a]Wall loss rate from Fry et al. (2009). [b]Wall loss rate from Perring et al. (2009). [c]Reaction rate constant from Martinez et al.
(1999). [d]Wall loss rates derived from the measurement of hydroxy nitrate wall loss.

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

Contributions of individual reactive biogenic volatile organic compounds to organic nitrates above
a mixed forest, Atmos. Chem. Phys., 12, 10125-10143, 10.5194/acp-12-10125-2012, 2012.
Presto, A. A., Hartz, K. E. H., and Donahue, N. M.: Secondary organic aerosol production from
terpene ozonolysis. 2. Effect of NOx concentration, Environ. Sci. Technol., 39, 7046-7054,
10.1021/es050400s, 2005.
Pye, H. O. T., Luecken, D. J., Xu, L., Boyd, C. M., Ng, N. L., Baker, K. R., Ayres, B. R., Bash, J.
O., Baumann, K., Carter, W. P. L., Edgerton, E. S., Fry, J. L., Hutzell, W. T., Schwede, D. B., and
Shepson, P. B.: Modeling the current and future roles of particulate organic nitrates in the
Southeastern United States, Environ. Sci. Technol., 49, 14195-14203, 10.1021/acs.est.5b03738,
882    2015.

Reinnig, M. C., Warnke, J., and Hoffmann, T.: Identification of organic hydroperoxides and
hydroperoxy acids in secondary organic aerosol formed during the ozonolysis of different
monoterpenes and sesquiterpenes by on-line analysis using atmospheric pressure chemical
ionization ion trap mass spectrometry, Rapid Commun. Mass Sp., 23, 1735-1741,
10.1002/rcm.4065, 2009.
Riipinen, I., Yli-Juuti, T., Pierce, J. R., Petaja, T., Worsnop, D. R., Kulmala, M., and Donahue, N.
M.: The contribution of organics to atmospheric nanoparticle growth, Nat. Geosci., 5, 453-458,
10.1038/ngeo1499, 2012.
Rindelaub, J. D., McAvey, K. M., and Shepson, P. B.: The photochemical production of organic
nitrates from alpha-pinene and loss via acid-dependent particle phase hydrolysis, Atmos. Environ.,
100, 193-201, 10.1016/j.atmosenv.2014.10.010, 2015.
Rindelaub, J. D., Borca, C. H., Hostetler, M. A., Slade, J. H., Lipton, M. A., Slipchenko, L. V.,
and Shepson, P. B.: The acid-catalyzed hydrolysis of an α-pinene-derived organic nitrate: kinetics,
products, reaction mechanisms, and atmospheric impact, Atmos. Chem. Phys., 16, 15425-15432,
10.5194/acp-16-15425-2016, 2016.
Rollins, A. W., Fry, J. L., Hunter, J. F., Kroll, J. H., Worsnop, D. R., Singaram, S. W., and Cohen,
R. C.: Elemental analysis of aerosol organic nitrates with electron ionization high-resolution mass
spectrometry, Atmos. Meas. Tech., 3, 301-310, 10.5194/amt-3-301-2010, 2010a.
Rollins, A. W., Smith, J. D., Wilson, K. R., and Cohen, R. C.: Real time in situ detection of organic
nitrates in atmospheric aerosols, Environ. Sci. Technol., 44, 5540-5545, 10.1021/es100926x,
2010b.
Rollins, A. W., Browne, E. C., Min, K. E., Pusede, S. E., Wooldridge, P. J., Gentner, D. R.,
Goldstein, A. H., Liu, S., Day, D. A., Russell, L. M., and Cohen, R. C.: Evidence for $NO_x$ control
over nighttime SOA formation, Science, 337, 10.1126/science.1221520, 2012.
Russell, L. M., Bahadur, R., and Ziemann, P. J.: Identifying organic aerosol sources by comparing
functional group composition in chamber and atmospheric particles, Proc. Nat. Acad. Sci., 108,
3516-3521, 10.1073/pnas.1006461108, 2011.
Saunders, S. M., Jenkin, M. E., Derwent, R. G., and Pilling, M. J.: Protocol for the development
of the Master Chemical Mechanism, MCM v3 (Part A): tropospheric degradation of non-aromatic
volatile organic compounds, Atmos. Chem. Phys., 3, 161-180, 10.5194/acp-3-161-2003, 2003.
Schwantes, R. H., Teng, A. P., Nguyen, T. B., Coggon, M. M., Crounse, J. D., St Clair, J. M.,
Zhang, X., Schilling, K. A., Seinfeld, J. H., and Wennberg, P. O.: Isoprene NO3 Oxidation
Products from the RO2 + HO2 Pathway, J. Phys. Chem. A, 119, 10158-10171,
10.1021/acs.jpca.5b06355, 2015.
Shepson, P. B., Mackay, E., and Muthuramu, K.: Henry's law constants and removal processes for
several atmospheric beta-hydroxy alkyl nitrates, Environ. Sci. Technol., 30, 3618-3623,
10.1021/Es960538y, 1996.
Shiraiwa, M., and Seinfeld, J. H.: Equilibration timescale of atmospheric secondary organic
aerosol partitioning, Geophys. Res. Lett., 39, 10.1029/2012gl054008, 2012.
Song, C., Na, K. S., and Cocker, D. R.: Impact of the hydrocarbon to NOx ratio on secondary
organic aerosol formation, Environ. Sci. Technol., 39, 3143-3149, 10.1021/es0493244, 2005.
Spittler, M., Barnes, I., Bejan, I., Brockmann, K. J., Benter, T., and Wirtz, K.: Reactions of NO3
radicals with limonene and alpha-pinene: Product and SOA formation, Atmos. Environ., 40, S116-
S127, 10.1016/j.atmosenv.2005.09.093, 2006.
Spracklen, D. V., Carslaw, K. S., Pöschl, U., Rap, A., and Forster, P. M.: Global cloud
condensation nuclei influenced by carbonaceous combustion aerosol, Atmos. Chem. Phys., 11,
9067-9087, 10.5194/acp-11-9067-2011, 2011.
Squire, O. J., Archibald, A. T., Griffiths, P. T., Jenkin, M. E., Smith, D., and Pyle, J. A.: Influence
of isoprene chemical mechanisms on modelled changes in tropospheric ozone due to climate and

land use over the 21st century, Atmos. Chem. Phys., 15, 5123-5143, 10.5194/acp-15-5123-2015, 2015.

Stocker, T. F., Qin, D., Plattner, G. K., Tignor, M., Allen, S. K., Boschung, J., Nauels, A., Xia, Y., Bex, V., and Midgley, P. M.: Climate Change 2013: the Physical Science Basis, contribution of Working Group 1 to the Fifth Assessment Report of the Intergovernmental Panel on Climate Change, Cambridge University Press, Cambridge, UK and New York, NY, USA, 2013.

Suda, S. R., Petters, M. D., Yeh, G. K., Strollo, C., Matsunaga, A., Faulhaber, A., Ziemann, P. J., Prenni, A. J., Carrico, C. M., Sullivan, R. C., and Kreidenweis, S. M.: Influence of functional groups on organic aerosol cloud condensation nucleus activity, Environ. Sci. Technol., 48, 10182-10190, 10.1021/es502147y, 2014.

Surratt, J. D., Gomez-Gonzalez, Y., Chan, A. W. H., Vermeylen, R., Shahgholi, M., and Kleindienst, T. E.: Organosulfate formation in biogenic secondary organic aerosol, J. Phys. Chem. A, 112, 8345-8378, 10.1021/jp802310p, 2008.

Tan, D., Faloona, I., Simpas, J. B., Brune, W., Shepson, P. B., Couch, T. L., Sumner, A. L., Carroll, M. A., Thornberry, T., Apel, E., Riemer, D., and Stockwell, W.: HOx budgets in a deciduous forest: Results from the PROPHET summer 1998 campaign, J. Geophys. Res.-Atmos., 106, 24407-24427, Doi 10.1029/2001jd900016, 2001.

Tsigaridis, K., and Kanakidou, M.: Secondary organic aerosol importance in the future atmosphere, Atmos. Environ., 41, 4682-4692, 10.1016/j.atmosenv.2007.03.045, 2007.

Valorso, R., Aumont, B., Camredon, M., Raventos-Duran, T., Mouchel-Vallon, C., Ng, N. L., Seinfeld, J. H., Lee-Taylor, J., and Madronich, S.: Explicit modelling of SOA formation from α-pinene photooxidation: sensitivity to vapour pressure estimation, Atmos. Chem. Phys., 11, 6895-6910, 10.5194/acp-11-6895-2011, 2011.

Vereecken, L., and Peeters, J.: Decomposition of substituted alkoxy radical-part I: A generalized structure-activity relationship for reaction barrier heights, Phys. Chem. Chem. Phys., 11, 9062-9074, 10.1039/B909712K, 2009.

von Schneidemesser, E., Monks, P. S., Allan, J. D., Bruhwiler, L., Forster, P., Fowler, D., Lauer, A., Morgan, W. T., Paasonen, P., Righi, M., Sindelarova, K., and Sutton, M. A.: Chemistry and the linkages between air quality and climate change, Chem. Rev., 115, 3856-3897, 10.1021/acs.chemrev.5b00089, 2015.

Wangberg, I., Barnes, I., and Becker, K.-H.: Product and mechanistic study of the reaction of NO3 radicals with alpha-pinene, Environ. Sci. Technol., 31, 2130-2135, 10.1021/es960958n, 1997.

Xiong, F., McAvey, K. M., Pratt, K. A., Groff, C. J., Hostetler, M. A., Lipton, M. A., Starn, T. K., Seeley, J. V., Bertman, S. B., Teng, A. P., Crounse, J. D., Nguyen, T. B., Wennberg, P. O., Misztal, P. K., Goldstein, A. H., Guenther, A. B., Koss, A. R., Olson, K. F., de Gouw, J. A., Baumann, K., Edgerton, E. S., Feiner, P. A., Zhang, L., Miller, D. O., Brune, W. H., and Shepson, P. B.: Observations of isoprene hydroxynitrates in the southeastern United States and implications for the fate of NO$_x$, Atmos. Chem. Phys., 15, 11257-11272, 10.5194/acp-15-11257-2015, 2015.

Xiong, F. L. Z., Borca, C. H., Slipchenko, L. V., and Shepson, P. B.: Photochemical degradation
of isoprene-derived 4,1-nitrooxy enal, Atmos. Chem. Phys., 16, 5595-5610, 10.5194/acp-16-5595-
972 2016, 2016.

Xu, L., Suresh, S., Guo, H., Weber, R. J., and Ng, N. L.: Aerosol characterization over the
southeastern United States using high-resolution aerosol mass spectrometry: spatial and seasonal
variation of aerosol composition and sources with a focus on organic nitrates, Atmos. Chem. Phys.,
15, 7307-7336, 10.5194/acp-15-7307-2015, 2015.
Yeh, G. K., and Ziemann, P. J.: Alkyl nitrate formation from the reactions of $C_8$-$C_{14}$ $n$-alkanes
with OH radicals in the presence of $NO_x$: measured yields with essential corrections for gas-wall
partitioning, J. Phys. Chem. A, 118, 8147-8157, 10.1021/jp500631v, 2014.
Zhang, X., Cappa, C. D., Jathar, S. H., Mcvay, R. C., Ensberg, J. J., Kleeman, M. J., and Seinfeld,
J. H.: Influence of vapor wall loss in laboratory chambers on yields of secondary organic aerosol,
Proc. Nat. Acad. Sci., 111, 5802-5807, 10.1073/pnas.1404727111, 2014.
Ziemann, P. J., and Atkinson, R.: Kinetics, products, and mechanisms of secondary organic aerosol
formation, Chem. Soc. Rev., 41, 6582-6605, 10.1039/C2CS35122F, 2012.
