# Peer review of "Nitrate radical oxidation of γ-terpinene: hydroxy nitrate, total organic nitrate, and"

_Atmospheric Chemistry and Physics, 2017_

## Referee Comment (RC1) · Anonymous Referee #1 · 24 Apr 2017

This paper presents a very interesting novel set of chamber experiments, examining the composition and SOA yield from NO3 + terpinene, and finding anomalously low organic nitrate and SOA yields compared to most monoterpenes but similar to that from α-pinene. This presents both an interesting mechanistic question – why are these yields so low? – which the authors address with some proposals, as well as providing important information for assessing overall OA budgets in regions where forest emissions may contain substantial terpinene, such as the upper Midwest of the U.S. The paper is very well written and the scientific conclusions are well supported in the text and figures. I recommend publication after addressing a few minor comments.

1) on p. 5 around line 85: Perhaps worth mentioning some other measurements of organic nitrate aerosol in the western US, by TD-LIF at BEACHON 2011 in the Colorado front range (http://www.atmos-chem-phys.net/13/8585/2013/), in various locations by FTIR (http://www.pnas.org/content/108/9/3516.full), and at Blodgett (http://www.atmos-chem-phys.net/12/5773/2012/acp-12-5773-2012.pdf). In the first two cases, they are surely likely to be multifunctional, and in the latter case the authors explicitly measure multifunctional nitrates. So I'd avoid the statement that these types of molecules have only been measured in the eastern US.

2) General comment: your chamber experiment likely has RO2 + NO3 (or RO2+RO2 where VOC » N2O5) as the dominant fate of the RO2 radical, which may bias to a particular product set. Given that ambient nighttime chemistry may have substantially more RO2 + HO2 reactions, maybe it's worth speculating on how that would affect your conclusions about paritioning implications.

3) p. 11 line 213-214, maybe reference Ng et al 2017 instead of Fry et al 2014, since Ng reviews all lit on NO3 + a-pinene. http://www.atmos-chem-phys.net/17/2103/2017/acp-17-2103-2017.pdf

4) p. 11, line 217 : "and likely g-terpinene, lose the nitrate moeity and hence are sufficiently volatile. . ."

5) p. 12: why would oxidation of 2nd double bond remove observed ON? Wouldn't this just double the amount of ON detected since you see each functional group and now there are 2 nitrate fxnal groups? The discussion of the secondary oxidation here is a bit confusing.

6) p. 14 VERY high Caero . So this makes your conclusions even more striking! Even at very high loading the yield was quite small.

7) p. 15 line 318 – could also be from RO2 + NO3

7.5) p. 15 line 365: I think there have recently been some hydroxynitrate quantifications during the FIXCIT chamber studies at Caltech, but to my knowledge none are published

yet.

8) p. 18 line 392: pinonaldehyde yield of 71% is not reported in Fry et al 2014. . . maybe wrong ref?

9) line 394: elaborate on "similar double bond character" – because each is adjacent to a branch point?

10) p. 19 around lines 404-407: does the MCM produce HO2 in your simulations of the a-pinene chemistry, via the mechanism you suggest, or any other?

11) p. 19, lines 415-417: Also assumes that only 1 double bond is reacted, right? Or do you include rates for each in your simple box model?

12) p. 20 lines 422-425: could split out as a stacked plot to highlight the sources of NO2, whether N2O5 dissociation or decomposition?

13) p. 23: is figure 1 wall-loss corrected?

14) Fig. 2: comment on uncertainty at low Mo – looks like SOA yield is not very well constrained that low.

15) line 491: "the right panel shows the same data on a log scale"

16) Fig. 4: could be clearer if same exponent on both scales – so the number scale could be different, but as it is now, both the scale and exponent are difference which makes it hard to see the slope. same for Fig. 5

17) Fig. 6: The O16 superscript in the label is a bit odd looking – necessary?

18) Fig. 9: related to 2 previous comments, could show modeled HO2 here, if it produces any. And could split out NO2 into recycled vs. N2O5 dissociation sources,

---

## Referee Comment (RC2) · Anonymous Referee #2 · 25 Apr 2017

Slade et al. present results from atmospheric simulation chamber experiments quantifying the gaseous ON yield, particulate ON yield and total SOA yield from the oxidation of gamma-terpinene (g-terpinene) by the NO3 radical. The work is thorough and contributes useful information for understanding the impact of ON formation on how and to what extent anthropogenic emissions can influence and are influenced by the oxidation of BVOCs. With some clarifications, this work deserves publication in ACP.

More discussion is needed on why no difference in yields was observed between seed and no seed experiments, particularly for the particle-phase ON yields. Current discussion (lines 253-264) is rather confusing. It is stated that under dry conditions, yields should be same with and without seed. Why? The results appear as if the yields are

indistinguishable with and without seed, yet the authors state, "However, ...." (line 260). Why would hydrolysis result in different yields with and without seed? What is the particle surface area available in each experiment? What is the surface area of the wall? My concern is that much of the multifunctional ON that would readily partition to the particle-phase is getting lost on chamber walls much faster than it is adsorbing to the particle surface, and that this is the reason why no significant difference in yields are observed. Gas-phase ON loss rate due to walls was determined using the signal for $C_{10}H_{17}NO_4$, which presumably would not partition to the particle-phase (or the walls) as fast as the multifunctional ON species. What is the gas-phase ON wall loss rate if $C_{10}H_{17}NO_5$ was used? Perhaps it may be useful to make a figure that shows the calculated wall loss rate versus O:C or oxygen atom number or whatever for each of the major ON species shown in figure 6. The entire discussion on ON_p yields is also muddied by the statement, "...signals close to the measured background noise..." (line 258). Please clarify.

Rindelaub et al. (2015) found hydrolysis of ON in the particle-phase can occur at low RH. Any evidence of hydrolysis during these experiments? Possible that with seed addition, some fraction of ON was lost due to hydrolysis that would not occur without seed?

Clarifications are needed for figure 6. What is the source of the signal between ~150 and ~320 m/z before NO3 addition? These signals do not vary with and without NO3 radical. Why? Why is there so much nitric acid in the no NO3 experiment? What is the large red signal left of the "O4" labeled peak? It is clearly enhanced by NO3. Why? Are all red signal > 320 m/z even mass? Was there no non-nitrate organic material detected with NO3 + gamma terpinene? Where is di-nitrate on this spectrum? The sum of all signal appear quite high relative to the reagent ion signal. What is iodide signal with nothing in the bag? Is the CIMS still linear with this much signal, that is, is it approaching reagent ion signal titration? Signals attributed to ON with 10 and 11 oxygen atoms highly questionable.

[Figure]

The discussion in the "atmospheric implication" section is inconsistent with the rest of the manuscript. These experiments were conducted near dry conditions. Discussion on mechanism focused solely on the higher volatility hydroxynitrates, not the multifunctional stuff (species with O=6, 7, 8 atoms in figure 6) that likely make up much of the ON mass in the particle-phase. Low yields all around were found. But the implication section still discusses what this may mean for SOA formation at high RH.

What was the relevance of ESI MS/MS to this work? Is the hydroxynitrate distinguishable from hydroperoxynitrate?

The detection of large signal that may be from a hydroperoxy nitrate is intriguing. Schwantes et al [2015 JPAa] also discussed this RO2+HO2 nighttime pathway (for isoprene). If the signal consistent with C10H17NO5 are in fact epoxides and not hydroperoxides, how would uptake differ with and without seed? What was pH of the seed particles?

To which alpha-pinene isomers of hydroxynitrate was the iodide CIMS calibrated? Include numbers for each isomer in table. Would you expect much variability from isomer to isomer? Would you expect much variability in the sensitivity to alpha-pine derived versus gamma-terpinene derived ON?

Minor Seed particles introduced into the chamber were ~100 nm. But the particle filter had pore size 1 micron.

What was the contribution of di-nitrates to total ON? How do you account for di-nitrates with FTIR? Can FTIR distinguish mono- from di-nitrates? How does this affect the way in which ON yield is calcualted?

"Teflon" (line 158) is trademark product that is similar to PTFE. So unless the filter is from Chemours that manufactures "Teflon", the appropriate description is PFA or PTFE or whatever.

---

## Referee Comment (RC3) · Anonymous Referee #3 · 27 Apr 2017

Overview The manuscript by Slade et al. presents new measurements on the oxidation of g-terpinene by the nitrate radical with a focus on understanding the organic nitrate yield. Both the gas- and particle-phase yields are reported as well as an SOA yield. A possible reaction mechanism is proposed based on observed products and structural similarities to other monoterpenes. Understanding organic nitrate yields is important for understanding NOx lifetime. Organic nitrate yields from monoterpene oxidation are currently poorly constrained thus limiting the community's ability to predict and understand NOx lifetime and reactive oxidized nitrogen partitioning in the atmosphere (particularly in biogenically influenced regions). The measurements presented in this paper are of interest to the community and will fill knowledge gaps. Prior to publication,

however, several areas need improvement/clarification.

General Comments: The introduction is lacking sufficient references to past work and in some instances misrepresents past work. For instance, the recent review by Ng et al. (2017) should be included. Additionally, multifunctional nitrates have been identified in locations other than the Eastern US (line 84). A few (non-comprehensive) examples include Beaver et al. (2012), Rollins et al. (2013), and Yan et al. (2016).

Some discussion on the fate of the peroxy radical and differences/similarities in between the chamber and the atmosphere is needed. In particular, it is likely that the $RO_2+HO_2$ is under-represented in the chamber experiments. How would this influence yields and the discussion in the "atmospheric implications" section? The model based on the master chemical mechanism could help inform this question.

Specific Comments: The in-line reference formatting is inconsistent and do not follow the journal standards. In particular, first initials are included in the in-line references for several authors.

Table 1: What is difference between the experiments on 11/12/15 and 11/18/15? They have similar deltaBVOC and ON yields but very different SOA yields. It would be useful to include the experiment length and the $NO_2$ concentration for each experiment as well. This could help explain some of the variability seen in Figures 3-5.

Figure 1: color scale missing

Figure 2: caption should say log scale (omit x-axis).

Figure 3/associated discussion lines 236-250: The data at deltaBVOC less than 300 ppb appears to be more scattered than the confidence interval might suggest. How much is this fit being influenced by the data points at high deltaBVOC? Or is some of the data at lower deltaBVOC influenced by more secondary oxidation? Some information on if the $NO_3$ concentrations were similar/very different would be very helpful here.

Figure 4: Please include the +2% slope as well that is stated on line 263.

Figure 6: Please label the hydroxynitrate peak used for the yield calculation. What is the source of the peaks from m/z 200-330 that appear in both experiments? Is the large peak at m/z 190 nitric acid? If so, what is the source in the absence of NO3? Are the O9, O10, and O11 signals real? This seem difficult to identify and are difficult to interpret in the figure in the current form. This figure may benefit from a zoomed in panel of the region of interest and specific identification of the ions discussed in the manuscript.

Lines 75-80: I find this explanation confusing and more elaboration is needed. The NO3 oxidation of both a-pinene and b-pinene lead to tertiary peroxy radicals.

Line 95: "...potential for ON and SOA formation are better understood." Better understood compared to what?

Line 133: Why not use the density of ammonium sulfate?

Line 245-246: Wouldn't some fraction of this yield dinitrates? How is that accounted for in this calculation?

Lines 256-258: This discussion should be elaborated on and clarified. The largest amount of particulate ON measured also corresponds to some of the largest scatter which seems somewhat inconsistent with the given explanation.

Line 317: should be "nitrooxyperoxy"

Line 408-409: Since a-pinene is being used as a surrogate for g-terpinene, I assume that the assumption is that only one of the double bonds is reacting. Is this correct or is further oxidation considered?

---

## Author Comment (AC1) · 2 Jun 2017

The comment was uploaded in the form of a supplement:
http://www.atmos-chem-phys-discuss.net/acp-2017-249/acp-2017-249-AC1-supplement.pdf

---

## Author Response (AR1)

DEPARTMENT OF CHEMISTRY

Dear Dr. Laskin,

In this document, we provide our revised manuscript, "Nitrate radical oxidation of γ-terpinene: hydroxy nitrate, total organic nitrate, and secondary organic aerosol yields", and a point-by-point response to the very careful and thoughtful critiques by the three anonymous referees, in the order the comments were raised. We believe we have satisfactorily responded to all referee comments, which has resulted in a significantly improved manuscript. While no changes were made to our original conclusions, the revised manuscript highlights and expands on relevant gas-phase $HO_2$ chemistry, secondary oxidation reactions, and potentially novel chemistry involving peroxyacyl nitrate epoxidation of olefins in the particle phase, further underscoring the importance of polyolefinic monoterpenes in SOA formation and chemistry. We believe the results presented in this manuscript will make a valuable contribution to the current literature and promote future studies on polyolefinic monoterpene oxidation. We hope you agree that our manuscript is in publishable form.

Our responses to each reviewer comment is provided below showing the reviewer comment in *italics* and reviewer response in normal font. Changes to the manuscript resulting from each reviewer comment are specified in each response to the reviewer by section and line number of the revised manuscript. Additions or modifications to the manuscript are indicated in bold font in the revised manuscript found at the bottom of this document.

We hope you agree that we have completely responded to all reviewer comments and that the paper is now in suitable form for publication. Thank you very much for your consideration.

Sincerely yours,

Jonathan H. Slade Jr., Ph.D.

*Department of Chemistry*

Brown Laboratory of Chemistry Building • 560 Oval Drive • West Lafayette, IN 47907-2084

**RESPONSES TO ANONYMOUS REVIEWER #1**
The authors greatly appreciate the comments and careful critique by reviewer #1, which have improved the quality of the manuscript. Below we provide a point-by-point response (shown in normal font) to the reviewer's comments (shown in italic font). Corresponding changes to the manuscript are shown in bold font of the revised version of our manuscript and indicated here based on line number.

**Reviewer comment #1:** *on p. 5 around line 85: Perhaps worth mentioning some other measurements of organic nitrate aerosol in the western US, by TD-LIF at BEACHON 2011 in the Colorado front range (http://www.atmos-chem-phys.net/13/8585/2013/), in various locations by FTIR (http://www.pnas.org/content/108/9/3516.full), and at Blodgett (http://www.atmoschem-phys.net/12/5773/2012/acp-12-5773-2012.pdf). In the first two cases, they are surely likely to be multifunctional, and in the latter case the authors explicitly measure multifunctional nitrates. So I'd avoid the statement that these types of molecules have only been measured in the eastern US.*

**Author response to reviewer comment #1:** We thank the reviewer for providing these references. They have been included in the revised version of the manuscript in lines 87-94.

**Reviewer comment #2:** *General comment: your chamber experiment likely has RO2 + NO3 (or RO2+RO2 where VOC » N2O5) as the dominant fate of the RO2 radical, which may bias to a particular product set. Given that ambient nighttime chemistry may have substantially more RO2 + HO2 reactions, maybe it's worth speculating on how that would affect your conclusions about partitioning implications.*

**Author response to reviewer comment #2:** We appreciate the reviewer's suggestion here as our chamber conditions may not represent conditions of all nighttime ambient forest environments. Given that ambient nighttime chemistry may have significantly more $RO_2+HO_2$ reactions, it is likely that the organic nitrate product distribution in ambient nighttime air is weighted more towards nitrooxy hydroperoxides. We have included a discussion of the impact of relative amounts of $HO_2$ between our chamber experiments and the ambient environment in sec. 3.3, lines 345-354 in the revised manuscript. We have also updated Fig. 9 (new Fig. 10) to show the evolution in the modeled ratio of $NO_3/HO_2$ and comparison to with ratios measured in the field.

**Reviewer comment #3:** *p. 11 line 213-214, maybe reference Ng et al 2017 instead of Fry et al 2014, since Ng reviews all lit on NO3 + a-pinene. http://www.atmos-chem-phys.net/17/2103/2017/acp- 17-2103-2017.pdf.*

**Author response to reviewer comment #3:** Thank you. We have instead referenced Ng et al., [2017].

**Reviewer comment #4:** *p. 11, line 217: "and likely g-terpinene, lose the nitrate moeity and hence are sufficiently volatile. . ."*

**Author response to reviewer comment #4:** This sentence has been modified accordingly in sec. 3.1, line 237 of the revised manuscript.

**Reviewer comment #5:** *p. 12: why would oxidation of 2nd double bond remove observed ON? Wouldn't this just double the amount of ON detected since you see each functional group and now there are 2 nitrate fxnal groups? The discussion of the secondary oxidation here is a bit confusing.*

**Author response to reviewer comment #5:** We thank the reviewer for highlighting this point. The expected first-generation oxidation products of γ-terpinene oxidation by $NO_3$ have a remaining double bond intact. Thus second-generation oxidation would likely occur at the position of the remaining double bond, potentially producing a second nitrate functionality, and therefore increase our FTIR-derived ON concentrations instead of decreasing them.

We have clarified in the methods section, lines 164-172, the potential effects of dinitrates on our derived ON concentrations, and sec. 3.2.1, lines 272-273 of the revised manuscript.

**Reviewer comment #6:** *p. 14 VERY high Caero. So this makes your conclusions even more striking! Even at very high loading the yield was quite small.*

**Author response to reviewer comment #6:** The authors agree with the reviewer's comment. Under these extremely high particle mass loadings, the SOA yields are quite low, e.g., compared to β-pinene+$NO_3$ as shown in Fig. 2 and discussed in the revised manuscript, sec. 3.1, lines 227-229.

**Reviewer comment #7:** *p. 15 line 318 – could also be from RO2 + NO3.*

**Author response to reviewer comment #7:** Since this section refers specifically to hydroxy nitrates, the $RO_2+NO_3$ pathway would lead to alkoxy radical formation and $NO_2$. It is unclear how this pathway would lead to hydroxy nitrate formation, therefore we have not made any changes to this particular line.

**Reviewer comment #7.5:** *p. 15 line 365: I think there have recently been some hydroxynitrate quantifications during the FIXCIT chamber studies at Caltech, but to my knowledge none are published yet.*

**Author response to reviewer comment #7.5:** To our knowledge, FIXCIT was designed around measurements of isoprene oxidation products and SOA. While we have included a reference to the FIXCIT campaign in the introduction, lines 92-93 of the revised manuscript, we feel the reference is not suited for this particular section of the manuscript.

**Reviewer comment #8:** *p. 18 line 392: pinonaldehyde yield of 71% is not reported in Fry et al 2014. . . maybe wrong ref?*

**Author response to reviewer comment #8:** The yield of 71% comes from the supplementary information of *Fry et al.*, [2014]. However, since the pinonaldehyde yield was not discussed in the main text of *Fry et al.*, [2014], we have removed the reference. In addition to the pinonaldehyde yield reported in *Wangberg et al.*, [1997], we now include the *Berndt and Böge* [1997] reference in sec. 3.5, lines 473-474 of the revised manuscript, which reports a pinonaldehyde yield of 75±6%.

**Reviewer comment #9:** *line 394: elaborate on "similar double bond character" – because each is adjacent to a branch point?*

**Author response to reviewer comment #9:** The reviewer is correct that when we refer to $\gamma$-terpinene as having similar double bond character as $\alpha$-pinene, we mean that both are adjacent to a branch point.

Clarification has been provided in sec. 3.5, lines 474-475 of the revised manuscript.

**Reviewer comment #10:** *p. 19 around lines 404-407: does the MCM produce HO2 in your simulations of the a-pinene chemistry, via the mechanism you suggest, or any other?*

**Author response to reviewer comment #10:** The reaction mechanism scheme, which is based on the MCM reaction scheme for $\alpha$-pinene+$NO_3$ as described in the supplementary material, does include $HO_2$ as one of the byproducts of keto-nitrate formation. The mechanism also includes consumption of $HO_2$ and formation of the hydroperoxy nitrates by the two $\alpha$- and $\beta$-nitrooxy peroxy radical isomers.

We have updated Fig. 9 (new Fig. 10), as requested in the reviewer's comment #18, to show the evolution of $HO_2$ based on the mechanism applied in the box model and added a discussion in sec. 3.5, lines 502-511 of the revised manuscript.

**Reviewer comment #11:** *p. 19, lines 415-417: Also assumes that only 1 double bond is reacted, right? Or do you include rates for each in your simple box model?*

**Author response to reviewer comment #11:** The *mechanism* assumes only one double bond is reacted, but the rate constant for $NO_3$ reaction is for that with $\gamma$-terpinene. This assumption does not affect the absolute concentration of the products. No changes have been made to the manuscript.

**Reviewer comment #12:** *p. 20 lines 422-425: could split out as a stacked plot to highlight the sources of NO2, whether N2O5 dissociation or decomposition?*

**Author response to reviewer comment #12:** We appreciate the suggestion here by the reviewer to clarify the contributions of $N_2O_5$ decomposition and recycling to the measured concentration of $NO_2$. However, since $NO_2$ is present in thermal equilibrium with $N_2O_5$ and only a small fraction is recycled following reaction (1-2%), we feel including a stacked plot doesn't add much value to Fig. 9 (new Fig. 10) or the manuscript. Besides removing the statement that the agreement between modeled and measured $NO_2$ concentrations in the chamber indicates reaction recycling, no other changes have been made.

**Reviewer comment #13:** *p. 23: is figure 1 wall-loss corrected?*

**Author response to reviewer comment #13:** Figure 1 was not corrected for wall loss. However, we have updated the figure to show wall loss-corrected SOA growth, consistent with our wall loss-corrected SOA yields.

**Reviewer comment #14:** *Fig. 2: comment on uncertainty at low Mo – looks like SOA yield is not very well constrained that low.*

**Author response to reviewer comment #14:** The authors agree with the reviewer. The modeled SOA yield is not very well constrained below an aerosol mass loading of ~30 $\mu$g m$^{-3}$. From the

95% confidence intervals, we now define the relative uncertainty in the yield at a mass loading of 10 μg m$^{-3}$ as +100/-50%.

We have modified the text in sec. 3.1, lines 221-224 of the revised manuscript.

**Reviewer comment #15:** *line 491: "the right panel shows the same data on a log scale"*

**Author response to reviewer comment #15:** Thank you for the suggested change, we have updated the Fig. 2 description as suggested by the reviewer.

**Reviewer comment #16:** *Fig. 4: could be clearer if same exponent on both scales – so the number scale could be different, but as it is now, both the scale and exponent are difference which makes it hard to see the slope. Same for Fig. 5*

**Author response to reviewer comment #16:** Thank you for the suggested changes. Figures 4 and 5 (new Fig. 6) have been updated as suggested by the reviewer.

**Reviewer comment #17:** *Fig. 6: The O16 superscript in the label is a bit odd looking – necessary?*

**Author response to reviewer comment #17:** Thank you. It has been removed (please refer to new Fig. 8).

**Reviewer comment #18:** *Fig. 9: related to 2 previous comments, could show modeled HO2 here, if it produces any. And could split out NO2 into recycled vs. N2O5 dissociation sources*

**Author response to reviewer comment #18:** Please refer to our response to reviewer comments #10 and #12.

**RESPONSES TO ANONYMOUS REVIEWER #2**

The authors greatly appreciate the thoughtful critique by reviewer #2, which has greatly improved the quality of the manuscript. Below we provide a point-by-point response (shown in normal font) to the reviewer's comments (shown in italic font). The resulting additions or alterations to the manuscript are indicated in bold font and provided in an updated version of the manuscript.

**Reviewer comment #1:** *More discussion is needed on why no difference in yields was observed between seed and no seed experiments, particularly for the particle-phase ON yields. Current discussion (lines 253-264) is rather confusing. It is stated that under dry conditions, yields should be same with and without seed. Why? The results appear as if the yields are indistinguishable with and without seed, yet the authors state, "However, ...." (line 260). Why would hydrolysis result in different yields with and without seed? What is the particle surface area available in each experiment? What is the surface area of the wall? My concern is that much of the multifunctional ON that would readily partition to the particle-phase is getting lost on chamber walls much faster than it is adsorbing to the particle surface, and that this is the reason why no significant difference in yields are observed. Gas-phase ON loss rate due to walls was determined using the signal for C10H17NO4, which presumably would not partition to the particle-phase (or the walls) as fast as the multifunctional ON species. What is the gas-phase ON wall loss rate if C10H17NO5 was used? Perhaps it may be useful to make a figure that shows the calculated wall loss rate versus O:C or oxygen atom number or whatever for each of the major ON species shown in figure 6. The entire discussion on ON_p yields is also muddied by the statement, "...signals close to the measured background noise..." (line 258). Please clarify.*

**Author response to reviewer comment #1:** We greatly appreciate the reviewer's insightful comments. The insignificant difference in the particle phase ON yields between the seeded and unseeded experiments may be due to the large fraction of organic material in the particles in both cases, and for the seeded experiments, relative to sulfate. During both the seeded and unseeded experiments, particle mass increased by orders of magnitude following uptake of the oxidation products. Thus, in terms of solubility, for example, the particles are essentially the same. Without sufficient liquid water and insignificant mass fractions of hygroscopic ammonium sulfate in the particles, reactions that remove ON, such as hydrolysis, likely play an insignificant role. To clarify, sec. 3.2.2, lines 282-288 have been modified accordingly.

Wall loss of the multifunctional ON products can be a concern in the derivation of both the gas and particle phase ON yields, particularly over long experimental timescales and when the particle surface area is low relative to the surface area of the chamber walls. The effects of semi-volatile wall loss on the reported gas phase ON yields are now included in the revised manuscript, sec. 3.2.1, lines 267-272. An extended discussion of the variability in particle-phase ON yields, including the effects of wall loss, has been included in sec. 3.2.2, lines 289-298 of the revised manuscript.

**Reviewer comment #2:** *Rindelaub et al. (2015) found hydrolysis of ON in the particle-phase can occur at low RH. Any evidence of hydrolysis during these experiments? Possible that with seed addition, some fraction of ON was lost due to hydrolysis that would not occur without seed?*

**Author response to reviewer comment #2:** Thank you. Please refer to the author response given to the reviewer's initial comment. It is possible that even at the low relative humidity in our experiments, some hydrolysis, catalyzed by the presence of the hygroscopic $(NH_4)_2SO_4$ seed and particle-phase $HNO_3$ could aid in the removal of the nitrate functionality (Rindelaub et al., 2015b). Given the virtually indistinguishable yields between the seeded and unseeded experiments, however, we believe that hydrolysis plays a very minor role in our experiments. We have added a brief discussion of the basis of this in sec. 3.2.2, lines 298-306, from two experiments conducted with added humidity. Regardless, we have accounted for these possible effects in the uncertainty of the particle-phase ON yield as described in the text.

**Reviewer comment #3:** *Clarifications are needed for figure 6. What is the source of the signal between ~150 and ~320 m/z before NO3 addition? These signals do not vary with and without NO3 radical. Why? Why is there so much nitric acid in the no NO3 experiment? What is the large red signal left of the "O4" labeled peak? It is clearly enhanced by NO3. Why? Are all red signal > 320 m/z even mass? Was there no non-nitrate organic material detected with NO3 + gamma terpinene? Where is di-nitrate on this spectrum? The sum of all signal appear quite high relative to the reagent ion signal. What is iodide signal with nothing in the bag? Is the CIMS still linear with this much signal, that is, is it approaching reagent ion signal titration? Signals attributed to ON with 10 and 11 oxygen atoms highly questionable*

**Author response to reviewer comment #3:** The source of the peaks in Fig. 6 (new Fig. 8) between $m/z$ ~150 and ~320 are associated with the background. Titration was not a concern as the reagent ion signal remained steady throughout the experiment, decreasing by ≤4% at the highest analyte concentrations. For clarity, we have updated Fig. 8 to show the relative enhancement over the background of several peaks in the presence of $NO_3$ and provide potential peak assignments for many of the unidentified peaks. While it is possible that the $O_{9-11}$ compounds are present in our system, we could not unambiguously make molecular assignments for them based on the mass spectra, and therefore, have removed them from Fig. 8.

Figure 6 (new Fig. 8) has been updated to reflect these changes and the discussion of this figure has been modified in sec. 3.4 lines 399-410.

**Reviewer comment #4:** *The discussion in the "atmospheric implication" section is inconsistent with the rest of the manuscript. These experiments were conducted near dry conditions. Discussion on mechanism focused solely on the higher volatility hydroxynitrates, not the multifunctional stuff (species with O=6, 7, 8 atoms in figure 6) that likely make up much of the ON mass in the particle-phase. Low yields all around were found. But the implication section still discusses what this may mean for SOA formation at high RH.*

**Author response to reviewer comment #4:** Thank you. We have added a section to the atmospheric implications regarding likely chemistry under atmospheric conditions, lines 521-523 and lines 530-536. A new figure (new Fig. 7) has been included, which reflects this chemistry.

**Reviewer comment #5:** *What was the relevance of ESI MS/MS to this work? Is the hydroxynitrate distinguishable from hydroperoxynitrate?*

**Author response to reviewer comment #5:** We appreciate the reviewer's comment as we felt we have not sufficiently highlighted the importance of our offline analysis of aerosol filter samples. While not used quantitatively, ESI MS/MS was applied to test for the presence of specific organic nitrate compounds and other oxidation products that we hypothesized were in the particle phase. This served, in part, as a means for support of the proposed mechanism shown in Fig. 8 (new Fig. 5). The hydroperoxy nitrate was distinguishable from the hydroxy nitrate based on its *m/z*. While we did not have a standard of the hydroperoxy nitrate to ensure the retention times matched that which was observed, we could, based on mass spectra and tandem mass spectra rankings, identify the chemical formula for each *m/z*.

We have provided clarification in the supplementary section.

**Reviewer comment #6:** *The detection of large signal that may be from a hydroperoxy nitrate is intriguing. Schwantes et al [2015 JPAa] also discussed this RO2+HO2 nighttime pathway (for isoprene). If the signal consistent with C10H17NO5 are in fact epoxides and not hydroperoxides, how would uptake differ with and without seed? What was pH of the seed particles?*

**Author response to reviewer comment #6:** The reviewer raises an interesting point regarding uptake of epoxides versus hydroperoxides as a function of seed and seed particle acidity. It is not entirely clear from a mechanistic standpoint how an epoxide would be produced in the gas phase from γ-terpinene oxidation by $NO_3$. Epoxides could, however, be produced in the condensed phase, for example, through a peroxyacyl nitrate (e.g. the corresponding PAN compound that might be produced from $NO_3$ reaction with terpinaldehyde), which could selectively epoxidize the remaining double bond on the hydroxy nitrate, as shown to be a very efficient epoxidation process by Darnall and Pitts (1970). Such a thermal reaction would be necessary for the dark chemistry studies here, as the isoprene epoxidation reported by Surratt et al (2010) requires OH oxidation of the precursor. Applying the Extended Aerosol Inorganics Model (E-AIM) (http://www.aim.env.uea.ac.uk/aim/aim.php), we estimate a pH ~5.5 for the $(NH_4)_2SO_4$ seed particles under saturated conditions becoming slightly more acidic as the relative humidity is decreased.

We have added a discussion in sec. 3.3, lines 354-368 of the revised manuscript regarding the pH of the particles, and the potential for condensed phase epoxidation and hydrolysis to produce species like a C10H17NO5 poly-ol, as shown in the new Fig.7.

**Reviewer comment #7:** *To which alpha-pinene isomers of hydroxynitrate was the iodide CIMS calibrated? Include numbers for each isomer in table. Would you expect much variability from isomer to isomer? Would you expect much variability in the sensitivity to alpha-pine derived versus gamma-terpinene derived ON?*

**Author response to reviewer comment #7:** We addressed this limitation in the original manuscript lines 354-363 and included the structure of the isomer in supplementary Fig. S4. The detailed synthesis and characterization of this isomer has been detailed in Rindelaub et al. (2016) and the supplementary information for that text. We have updated the text where applicable to clarify that we calibrate for only one isomer of α-pinene hydroxy nitrate.

**Reviewer comment #8:** *Seed particles introduced into the chamber were ∼100 nm. But the particle filter had pore size 1 micron.*

**Author response to reviewer comment #8:** The filters were 1 μm pore size polytetrafluoroethylene (PTFE), which boasts a collection efficiency for particles between 0.01 μm and 1 μm of ~100% (Burton et al., 2007). We have updated the methods section in lines 178-176 of the revised manuscript.

**Reviewer comment #9:** *What was the contribution of di-nitrates to total ON? How do you account for di-nitrates with FTIR? Can FTIR distinguish mono- from di-nitrates? How does this affect the way in which ON yield is calcualted?*

**Author response to reviewer comment #9:** This is a good question. We did not differentiate mono- from di-nitrates using FTIR as the asymmetric -$NO_2$ stretch is not specific to any particular organic nitrate but all organic nitrates (Roberts, 1990). This could result in an overestimation in the reported ON yields as di-nitrates are expected to absorb more than the mono-nitrates in the wavelength region of interest. However, given the relative rate constants of alkoxy radical reaction with $NO_2$ (~$10^{-11}$ $cm^3$ $s^{-1}$) and $O_2$ (~$10^{-14}$ $cm^3$ $s^{-1}$) (Atkinson et al., 1982) and the relative concentrations of $NO_2$ (~$8\times10^{12}$ $cm^{-3}$ maximum) and $O_2$ (~$5\times10^{18}$ $cm^{-3}$) in the chamber, we expect an insignificant contribution from first-generation di-nitrates (<0.2%). Some contribution of secondary oxidation to di-nitrates (~10%) is possible, based on the relative rates of first-generation to second-generation monoterpene oxidation.

We have updated the Methods section in lines 164-172 of the revised manuscript to address the potential influence of di-nitrates on the reported ON yields.

**Reviewer comment #10:** *"Teflon" (line 158) is trademark product that is similar to PTFE. So unless the filter is from Chemours that manufactures "Teflon", the appropriate description is PFA or PTFE or whatever*

**Author response to reviewer comment #10:** Where applicable, we have removed "Teflon" and replaced with PTFE or PFA.

**RESPONSES TO ANONYMOUS REVIEWER #3**
The authors greatly appreciate the careful review by reviewer #3. Below we provide a point-by-point response (shown in normal font) to the reviewer's comments (shown in italic font). The resulting additions or alterations to the manuscript are indicated in bold font in an updated version of the manuscript.

**Reviewer comment #1:** *The introduction is lacking sufficient references to past work and in some instances misrepresents past work. For instance, the recent review by Ng et al. (2017) should be included. Additionally, multifunctional nitrates have been identified in locations other than the Eastern US (line 84). A few (non-comprehensive) examples include Beaver et al. (2012), Rollins et al. (2013), and Yan et al. (2016).*

**Author response to reviewer comment #1:** Thank you. See our response to the first comment from reviewer #1. We have updated the introduction to include several other locations where multifunctional organic nitrates have been measured.

**Reviewer comment #2:** *Some discussion on the fate of the peroxy radical and differences/similarities in between the chamber and the atmosphere is needed. In particular, it is likely that the RO2+HO2 is under-represented in the chamber experiments. How would this influence yields and the discussion in the "atmospheric implications" section? The model based on the master chemical mechanism could help inform this question.*

**Author response to reviewer comment #2:** This is a good question and similar to the second comment raised by reviewer #1. For more details, please refer to the author response given to reviewer #1 and updated Fig. 9 (new Fig. 10). In short, we have updated the section on aerosol partitioning to address how differences in $HO_2$ between our experiments and the ambient environment affects interpretation of the SOA yield and formation of organic hydroperoxides. Moreover, Fig. 10 has been updated to show modeled $HO_2$ and $NO_3/HO_2$ ratios in the reaction chamber.

**Reviewer comment #3:** *The in-line reference formatting is inconsistent and do not follow the journal standards. In particular, first initials are included in the in-line references for several authors.*

**Author response to reviewer comment #3:** We have fixed this typo in the revised manuscript.

**Reviewer comment #4:** *Table 1: What is difference between the experiments on 11/12/15 and 11/18/15? They have similar deltaBVOC and ON yields but very different SOA yields. It would be useful to include the experiment length and the NO2 concentration for each experiment as well. This could help explain some of the variability seen in Figures 3-5.*

**Author response to reviewer comment #4:** We have updated table 1 to include the experimental time and concentration of $NO_2$ when it was measured. Considering the significant difference in $NO_2$ concentrations for these two periods, under high $NO_x$ conditions (11/12/15 experiment), new particle formation or aerosol growth may be suppressed (lower SOA yield) due to competitive chemistry of peroxy radicals between $NO_3$ and $HO_2$, with the $RO_2+NO_3$ pathway leading to more volatile reaction products. Similar observations have been made, for example, during the ozonolysis of α-pinene under high $NO_x$ conditions (Presto et al., 2005).

We have modified the sec. 3.3 heading and provided discussion of the influence of high $NO_3$ concentrations and the fate of $RO_2$ in sec. 3.3, lines 368-375 of the revised manuscript.

**Reviewer comment #5:** *Figure 1: color scale missing*

**Author response to reviewer comment #5:** Figure 1 has been updated to include color scale.

**Reviewer comment #6:** *Figure 2: caption should say log scale (omit x-axis).*

**Author response to reviewer comment #6:** We have removed "x axis" in the caption.

**Reviewer comment #7:** *Figure 3/associated discussion lines 236-250: The data at deltaBVOC less than 300 ppb appears to be more scattered than the confidence interval might suggest. How much is this fit being influenced by the data points at high deltaBVOC? Or is some of the data at lower deltaBVOC influenced by more secondary oxidation? Some information on if the NO3 concentrations were similar/very different would be very helpful here.*

**Author response to reviewer comment #7:** Thank you for the suggestion. The reviewer is correct that the 95% confidence intervals of the fit do not capture the variability, e.g., at $\Delta BVOC \sim 200$ ppb, and the data points above $\Delta BVOC \sim 300$ ppb do lead to an enhancement in the yield. The exact cause of the data variability below $\Delta BVOC \sim 300$ ppb is not entirely clear, but may be a combination of larger relative uncertainties in $\Delta ON$ and $\Delta BVOC$ at lower $\Delta BVOC$ and different $NO_3$ concentrations and experimental timescales.

We have included a discussion of the potential causes of the variability in Fig. 3 in sec. 3.2.1, lines 264-273 of the revised manuscript.

**Reviewer comment #8:** *Figure 4: Please include the +2% slope as well that is stated on line 263.*

**Author response to reviewer comment #8:** We have modified the slopes presented in Figs. 4, 5 (new Fig. 6) and 7 (new Fig. 9), to reflect those presented in the main text.

**Reviewer comment #9:** *Figure 6: Please label the hydroxynitrate peak used for the yield calculation. What is the source of the peaks from m/z 200-330 that appear in both experiments? Is the large peak at m/z 190 nitric acid? If so, what is the source in the absence of NO3? Are the O9, O10, and O11 signals real? This seem difficult to identify and are difficult to interpret in the figure in the current form. This figure may benefit from a zoomed in panel of the region of interest and specific identification of the ions discussed in the manuscript.*

**Author response to reviewer comment #9:** Similar comments were made by reviewer #2. Please see our response to their comment #3 for more details. Briefly, the signals between *m/z* 200 and 330 are attributed to the background. We include a new trace in Fig. 6 (shown in black in new Fig. 8) that shows the enhancement over the background following addition of $NO_3$ to the chamber. We have removed the $O_{9-11}$ assignments as they could not be unambiguously verified. The hydroxy nitrate peak used for the yield calculation is now stated in the Fig. 8 description.

**Reviewer comment #10:** *Lines 75-80: I find this explanation confusing and more elaboration is needed. The NO3 oxidation of both a-pinene and b-pinene lead to tertiary peroxy radicals.*

**Author response to reviewer comment #10:** The authors agree with the reviewer here that this statement is too general and does not fully explain the differences in yields observed for monoterpenes that also lead to tertiary peroxy radical formation (Fry et al., 2014).

We have provided clarification in the Introduction section, lines 78-84 of the revised manuscript.

**Reviewer comment #11:** *Line 95: ". . .potential for ON and SOA formation are better understood." Better understood compared to what?*

**Author response to reviewer comment #11:** This should instead read "…better studied compared to other monoterpenes…". Lines 105-106 of the revised manuscript reflect this change.

**Reviewer comment #12:** *Line 133: Why not use the density of ammonium sulfate?*

**Author response to reviewer comment #12:** The reviewer is correct that for the seeded experiments, initial particle density should be that of $(NH_4)_2SO_4$ of ~1.7 g cm$^{-3}$. We have recalculated the initial seed particle mass concentration based on a density of 1.7 g cm$^{-3}$. Leaving the density of the SOA as 1.2 g cm$^{-3}$ has resulted in a slight but insignificant decrease in $\Delta M$ and SOA yields for the $(NH_4)_2SO_4$ seeded experiments as presented in Table 1 and Fig. 2.

**Reviewer comment #13:** *Line 245-246: Wouldn't some fraction of this yield dinitrates? How is that accounted for in this calculation?*

**Author response to reviewer comment #13:** We have clarified this limitation in the revised manuscript as detailed in our response to comment #9 by reviewer #2.

**Reviewer comment #14:** *Lines 256-258: This discussion should be elaborated on and clarified. The largest amount of particulate ON measured also corresponds to some of the largest scatter which seems somewhat inconsistent with the given explanation.*

**Author response to reviewer comment #14:** For more details, please see comments #1 and #2 by reviewer #2 and our response to comment #7 here. In short, we have updated the discussion to include the potential influence of hydrolysis, secondary oxidation, and different $RO_2$ loss pathways at variable $NO_3$ concentrations in the chamber.

**Reviewer comment #15:** *Line 317: should be "nitrooxyperoxy"*

**Author response to reviewer comment #15:** Thank you, we have fixed this typo.

**Reviewer comment #16:** *Line 408-409: Since a-pinene is being used as a surrogate for g-terpinene, I assume that the assumption is that only one of the double bonds is reacting. Is this correct or is further oxidation considered?*

**Author response to reviewer comment #15:** The reviewer is correct, we assume that only one of the double bonds of γ-terpinene reacts with $NO_3$. Please refer to comment #11 by reviewer #1 for more detail.

[revised manuscript text omitted]